# Partitioned polygenic scores show mechanistic heterogeneity in type 2 diabetes and hypertension comorbidity

Vincent Pascat [1,2], Liudmila Zudina[2,3], Lucas Maurin[1], Anna Ulrich [2], Jared G. Maina[1], Ayse Demirkan [2,3,4], Zhanna Balkhiyarova [2,3,4], Igor Pupko[3], Yevheniya Sharhorodska [3,5], François Pattou [1,6], Bart Staels [7], Marika Kaakinen [2,3,4,8], Amna Khamis[1,2], Amélie Bonnefond [1,2], Patricia Munroe [9,10], Philippe Froguel [1,2,11] & Inga Prokopenko [1,3,4,11] ✉

Type 2 diabetes and hypertension are common health conditions that often occur together, suggesting shared biological mechanisms. To explore this relationship, we analyse large-scale multiomic data to uncover genetic factors underlying type 2 diabetes and blood pressure comorbidity. We curate 1304 independent single-nucleotide variants associated with type 2 diabetes and blood pressure, grouping them into five clusters related to metabolic syndrome, inverse type 2 diabetes/blood pressure risk, impaired pancreatic beta-cell function, higher adiposity, and vascular dysfunction. Colocalization with tissue-specific gene expression highlights significant enrichment in pathways related to thyroid function and fetal development. Partitioned polygenic scores derived from these clusters improve risk prediction for type 2 diabetes/hypertension comorbidity, identifying individuals with more than twice the usual susceptibility. These results reveal a mechanistically heterogeneous genetic architecture shared between type 2 diabetes and blood pressure, enhancing comorbidity risk prediction. Partitioned polygenic risk scores offer a promising approach for early risk stratification, personalised prevention, and improved management of these interconnected conditions.

Hypertension and type 2 diabetes (T2D) pose major public health challenges, affecting approximately 1.28 billion[1] and 537 million adults worldwide[2], respectively, with the prevalence of T2D expected to rise to 1.3 billion by 2050. T2D and high blood pressure (BP) frequently co-occur in the same individual[3–5]. The T2D-BP comorbidity further increases the risk of major health outcomes, and individuals with both conditions often face challenges in achieving treatment objectives[6]. T2D and high BP are key components of the Metabolic Syndrome (MetS), also encompassing various cardiovascular risk factors, including central obesity, dyslipidaemia, microalbuminuria, and insulin resistance (IR)[7,8].

[1]Université de Lille, Inserm UMR1283, CNRS UMR8199, European Genomic Institute for Diabetes (EGID), Institut Pasteur de Lille, Lille University Hospital, Lille, France. [2]Department of Metabolism, Digestion, and Reproduction, Imperial College London, London, UK. [3]Section of Statistical Multi-omics, Department of Clinical and Experimental Medicine, University of Surrey, Guildford, UK. [4]People-Centred Artificial Intelligence Institute, University of Surrey, Guildford, UK. [5]Department of Life Sciences and Biotechnology, University of Ferrara, Ferrara, Italy. [6]Department of General and Endocrine Surgery, CHU Lille, Lille, France. [7]Université de Lille, Inserm, CHU Lille, Institut Pasteur de Lille, U1011-EGID, Lille, France. [8]Institute for Molecular Medicine Finland, University of Helsinki, Helsinki, Finland. [9]William Harvey Research Institute, Barts and the London Faculty of Medicine and Dentistry, Queen Mary University of London, London, UK. [10]National Institute of Health and Care Research, Barts Cardiovascular Biomedical Research Centre, Queen Mary University of London, London, UK. [11]These authors jointly supervised this work: Philippe Froguel, Inga Prokopenko. ✉e-mail: i.prokopenko@surrey.ac.uk

Extensive genetic research, notably through recent genome-wide association studies (GWAS), has dissected the underlying genetic architecture of both T2D and BP traits independently. Latest reports associated 1289 independent variants in DNA with T2D[9], while 2103 variants are implicated in BP control[10]. These findings highlight the complex genetic architecture of T2D and BP traits/hypertension, emphasising their diverse genetic drivers.

Despite significant advances in understanding the genetics of T2D and high BP as independent conditions, the shared genetic basis underlying their frequent comorbidity remains largely unexplored. This gap persists even though T2D–high BP comorbidity has been consistently observed, including within genetic datasets[11–13]. Several Mendelian randomisation (MR) studies have yielded conflicting evidence about the causal relationship between T2D and high BP. For instance, Sun et al. identified T2D as a driver of high BP, while Aikens et al. proposed the opposite[14,15]. Additionally, another study found that two of four types of hypertensive medications were protective against T2D risk, while the others increased its risk[16]. These findings highlight the diverse and complex pathways underpinnings the T2D-BP relationship.

In this study, we aimed to characterise the shared pathophysiological processes underlying the T2D-BP relationship by harnessing large-scale genomic datasets from recent research on both conditions. Using common genetic variation, we sought to enhance the mechanistic understanding of these diseases comorbid status and suggest potential avenues for targeted interventions and precision health.

We aggregated genomic data from 45 GWAS for related conditions and traits/endophenotypes, 50 tissue-specific expression quantitative trait *loci* (eQTL)[17–19], assay for transposase-accessible chromatin using sequencing peaks from single-cell (scATAC-seq) atlas[20], and the UK Biobank (UKB) cohort[21]. By leveraging GWAS summary statistics, we assessed the genetic correlation between T2D[22] and systolic BP (SBP), diastolic BP (DBP), and pulse pressure (PP = SBP-DBP)[12]. We clustered the T2D-BP-associated independent single-nucleotide variant (SNV) effects into distinct groups based on their underlying pathogenetic processes. We observed the cluster-associated changes in gene expression through colocalization analysis with eQTL and enrichment in the scATAC-seq atlas. We finally evaluated the cluster-specific risks of complication using partitioned polygenic scores (PGS) in 459,247 individuals (Fig. 1).

## Results

### Genetic overlap between T2D and BP

We explored the genetic relationships between T2D and BP by evaluating the overall genetic correlation and associated *loci* that overlap between the two conditions. We performed linkage disequilibrium (LD) score regression using *ldsc*[23] and observed a direct genetic correlation between T2D and SBP ($r_g$[SE] = 0.25[0.028], $p = 1.56 \times 10^{-19}$), DBP ($r_g$[SE] = 0.18[0.027], $p = 1.38 \times 10^{-11}$), and PP ($r_g$[SE] = 0.23[0.029], $p = 2.25 \times 10^{-15}$), consistent with previous research.

To further validate the LD score regression results, we constructed PGSs for T2D, SBP, DBP and PP in the UKB (Supplementary Table 1) using independent weights from GWASs ("Methods"). We probed whether a genetic predisposition towards one condition could predict the risk of the other using *comorbidPGS*[24]. T2D PGS was consistently associated with a modest increase in SBP, DBP, and PP (Beta_PP[SE] ≥ 0.37[0.017] change in PP mmHg per one-unit increase in T2D PGS, $p \le 1.83 \times 10^{-106}$). SBP and PP PGSs were significantly associated with a higher risk of T2D (OR_SBP[95% CI] = 1.07 [1.06–1.09] change in T2D odds per one-unit increase in SBP PGS, $p = 9.36 \times 10^{-35}$; OR_PP[95% CI] = 1.07 [1.06–1.08], $p = 8.02 \times 10^{-31}$). In contrast, DBP PGS had no impact on T2D risk (OR_DBP[95% CI] = 1.01[0.999–1.02], $p = 0.062$, Supplementary Table 2).

We gathered a collection of 1401 SNVs associated with T2D, high SBP, DBP, and/or PP ("Methods"). We revealed 24/19/26

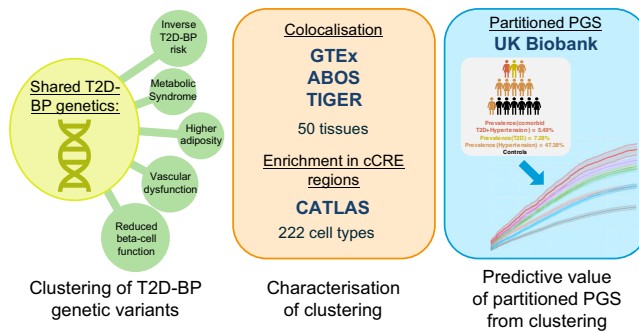

**Fig. 1 | Study overview.** The genetic landscape of T2D-BP is explored and subsequently clustered into five groups of different pathogenetic processes. These clusters are then analysed using colocalization methods, eQTL and the scATAC-seq atlas to identify associated changes in gene expression. The clusters are evaluated for their predictive value and the underlying causal mechanisms contributing to the T2D-BP relationship. T2D type 2 diabetes, BP blood pressure, cCRE candidate cis-regulatory elements, PGS polygenic scores, DBP diastolic blood pressure, SBP systolic blood pressure.

overlapping genetic *loci* between T2D and SBP/DBP/PP, respectively, determined by LD and/or genomic proximity (Supplementary Data 1). Of the 1401 SNVs, 9 were directly reported as lead signals for both T2D and BP traits. Additionally, we identified 97 SNV pairs that overlapped (within 500 kb or LD $r^2 > 0.2$) and were associated with T2D and BP traits. We observed several well-known *loci*, such as those at *GRB14-COBLL1* (associated with reduced insulin level, pulse pressure, and mean arterial pressure)[25,26], *ADCY5* (beta cell function and lipodystrophy)[27], and *ACE* (renin-angiotensin system, hypertension) genes[28]. Overlapping *loci* at *JAZF1* (regulating glucose, lipid, and inflammation)[29], *ADRB1* (beta-adrenergic receptors regulating cardiac contractility and heart rate)[30], *TCF7L2* (controlling Langerhans islet proliferation)[31], and *SGIP1* (signalling in energy homoeostasis)[32] contribute to the inverse relationship between T2D and BP. These results highlight the dense and complex genetic relationships between high BP predisposition and T2D risk.

### Clusters of pathogenetic processes

To dissect the complexity of shared biological pathways between T2D and BP, we curated and refined the SNV list to 1304 independent variants (LD $r^2 < 0.2$), including 500 T2D-associated and 813 BP-associated SNVs, to cluster them based on their effects on 45 related (endo) phenotypes, including T2D/BP traits (Supplementary Data 2). Our investigation encompassed a wide array of related endpoints or risk factors, including biomarkers of inflammation and hepatic function, circulating plasma lipids, cardiovascular health indicators, anthropometric measures, glycaemic traits, and sex hormones (Supplementary Data 2). All SNVs (originally associated with T2D, BP traits, or both) were aligned to the T2D risk allele. When information for a particular phenotype at an SNV was unavailable, we used LD proxies ("Methods")[33], and performed imputation of the remaining missing data by random forest algorithm implemented in the *imputeSCOPA* software tool[34]. We used an unsupervised hierarchical clustering approach, given the anticipated heterogeneity within our SNV set induced by the inherent complexity in both T2D and BP signals[35]. To ensure robustness of our clustering, we ran extensive sensitivity analyses ("Methods") with different sets of metabolic traits and other clustering methods, such as using Z-score adjusted for GWAS sample size, *MRClust*[36] and Bayesian nonnegative matrix factorization (bNMF) ("Methods", Supplementary Figs. 1–4). We identified five clusters of distinct pathogenetic mechanisms (Fig. 2), highlighting mechanistic heterogeneity in T2D-BP comorbidity. We compared the SNV assignments of our T2D-BP clusters with recent T2D hierarchical clustering[9]

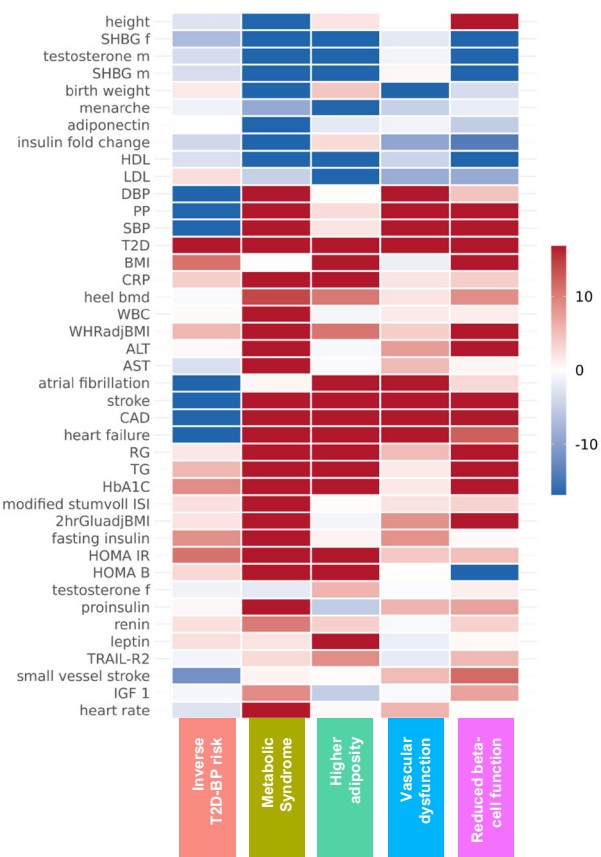

**Fig. 2 | Clustering heat map of endophenotypes with five pathogenetic SNV clusters associated with high BP and/or risk of T2D.** Each row corresponds to a GWAS of an endophenotype, while each column corresponds to a cluster (Supplementary Data 4). Colour represents the direction of the *z*-score (aligned to the T2D risk allele) from two-sided association tests between the GWAS and SNVs assigned to each cluster. Colour intensity represents the significance, expressed as −log(*p* value) from two-sided *t*-tests on the regression coefficients linking SNV cluster assignment with GWAS effect sizes. T2D type 2 diabetes, DBP diastolic blood pressure, UKB UK Biobank, PP pulse pressure, SBP systolic blood pressure, HbA1C glycated haemoglobin, RG random glucose, WHR waist-hip ratio, BMI body mass index, HDL high-density lipoprotein, PAI plasminogen activator inhibitor, ISI insulin sensitivity index, IGF insulin-like growth factor, LDL low-density cholesterol, adjBMI adjusted for BMI, HOMA homoeostatic model assessment, IR insulin resistance, B beta-cell function, WBC white blood cell count, CRP C-reactive protein, CAD coronary artery disease, TG triglycerides, SHBG sex-hormone-binding globulin, TRAIL-R2 TNF related apoptosis inducing ligand receptor 2, ALT alanine aminotransferase, AST aspartate aminotransferase, BMD bone mass density.

and bNMF clustering[37,38] ("Methods", Supplementary Data 3, 4, Supplementary Figs. 5, 6). The pathophysiological processes identified across the five clusters were consistent with existing evidence and highlight mechanistic insights ("Methods", Supplementary Figs. 4–7)[9,12,37].

The *Metabolic Syndrome* cluster included 215 variants, and displayed the most distinct pathogenetic signature. It highlights attributes consistent with the metabolic syndrome, including lower levels of sex hormones (sex-hormone binding globulin, insulin, testosterone)[39,40], higher central adiposity (waist-to-hip ratio [WHR] adjusted for body-mass index [BMI])[41] without higher overall adiposity, measured by BMI, systemic higher IR evaluated by the homoeostasis model assessment of insulin resistance, HOMA-IR, using both fasting plasma glucose and insulin (alongside higher HOMA-B, proinsulin level, and insulin fold change), lower high-density lipoprotein (HDL) cholesterol, higher triglycerides (TG), and altered cardiovascular functions (higher heart rate, increased cardiovascular event risk,

higher renin-angiotensin-aldosterone system activity)[8]. SNVs within this cluster also strongly associate with shorter stature (lower height) and lower birth weight.

Previous findings reported that shorter stature is associated with a higher risk of T2D[42] and cardiovascular events[38,43]. Other studies linked greater height with insulin and insulin-like growth factor signalling pathways[44]. The impaired insulin sensitivity may be one of the underlying factors in this association[45,46].

The T2D–high BP comorbidity is high in this cluster and was consistent with our bNMF clustering (Supplementary Fig. 3). The origin of the SNVs is an equal mix of T2D and BP (Supplementary Fig. 1). When comparing SNVs with previous T2D clustering, we observed an overlap between the SNVs in our *Metabolic Syndrome* cluster and the Type 2 Diabetes Global Genetics Initiative (T2DGGI) *Metabolic syndrome* cluster, as well as the bNMF cluster *Lipodystrophy 1* (Supplementary Fig. 5).

In the *Inverse T2D-BP risk* cluster, we noted an inverse relationship of associated SNVs effects on higher T2D risk related to lower SBP/DBP/PP. Predominantly originating from associations with BP traits (Supplementary Fig. 1), the 353 SNVs within this cluster, when aligned to the T2D risk allele, are associated with a lower risk of cardiovascular events, such as atrial fibrillation (AF), coronary artery disease (CAD), stroke and heart failure. Additionally, these SNVs demonstrated associations with BMI and systemic higher IR (higher HOMA-IR). Comparison with previously reported BP-related clusters showed partial overlap with the *Hypolipidaemia* and *Short stature* SNV groups (Supplementary Fig. 6)[38].

The *Higher adiposity* cluster contained 137 SNVs—predominantly T2D signals—which showcased effects on higher BMI, reduced sex hormones (testosterone and SHBG), higher TG along with lower HDL- and LDL-cholesterol, higher risk of cardiovascular events (CAD, heart rate, stroke), and insulin resistance (higher HOMA-IR/HOMA-B). This cluster, distinct from the *Metabolic Syndrome* one, showed a high number of obesity-related SNVs within previous T2D clustering (Supplementary Fig. 5). The *Vascular Dysfunction* cluster included 287 SNVs mostly originating as BP signals. They are associated with cardiovascular traits (higher risk of AF, stroke, CAD, heart failure), lower birth weight and show strong effects on both T2D–high BP. This cluster showed a number of hypolipidemia SNVs from previous BP clustering (Supplementary Fig. 6). Lastly, *Reduced beta-cell function* cluster exhibited characteristics of impaired beta-cell function including lower homoeostasis model assessment of beta cell function (HOMA-B), higher glucose/glycated haemoglobin levels (random glucose [RG], HbA1c), metabolic dysregulation (TG, sex hormones), higher inflammation (C-reactive protein [CRP], IGF-1) and taller stature (height). The *Reduced beta-cell function* cluster contained 312 SNVs, predominantly T2D signals, found in the *Beta cell 1* and *Beta cell 2* clusters from the latest published T2D bNMF clustering (Supplementary Fig. 5b).

The five distinct mechanistic groups of genetic variants, revealed through clustering, contribute to the shared susceptibility to T2D and high BP. They provide a foundation for further exploration of the biological pathways.

## Multiomic characterisation of T2D-BP clusters

To further characterise the T2D-BP comorbidity clusters, we evaluated the changes in gene expression and regulatory elements associated with the clustered SNVs. We first conducted a colocalization analysis to elucidate the impact of studied SNVs on gene expression patterns. We explored the genomic landscape within a 200 kb window surrounding each clustered SNV to assess the likelihood of a shared causal variant between our clusters and gene expression changes across 50 tissues from various eQTLs datasets using a hypothesis-free approach and the *coloc* R package ("Methods")[47]. We identified a total of 6321 colocalizations across the 50 tissues, involving 1558 genes and 448 clustered variants (Fig. 3a top, Supplementary Data 5, Supplementary Fig. 8).

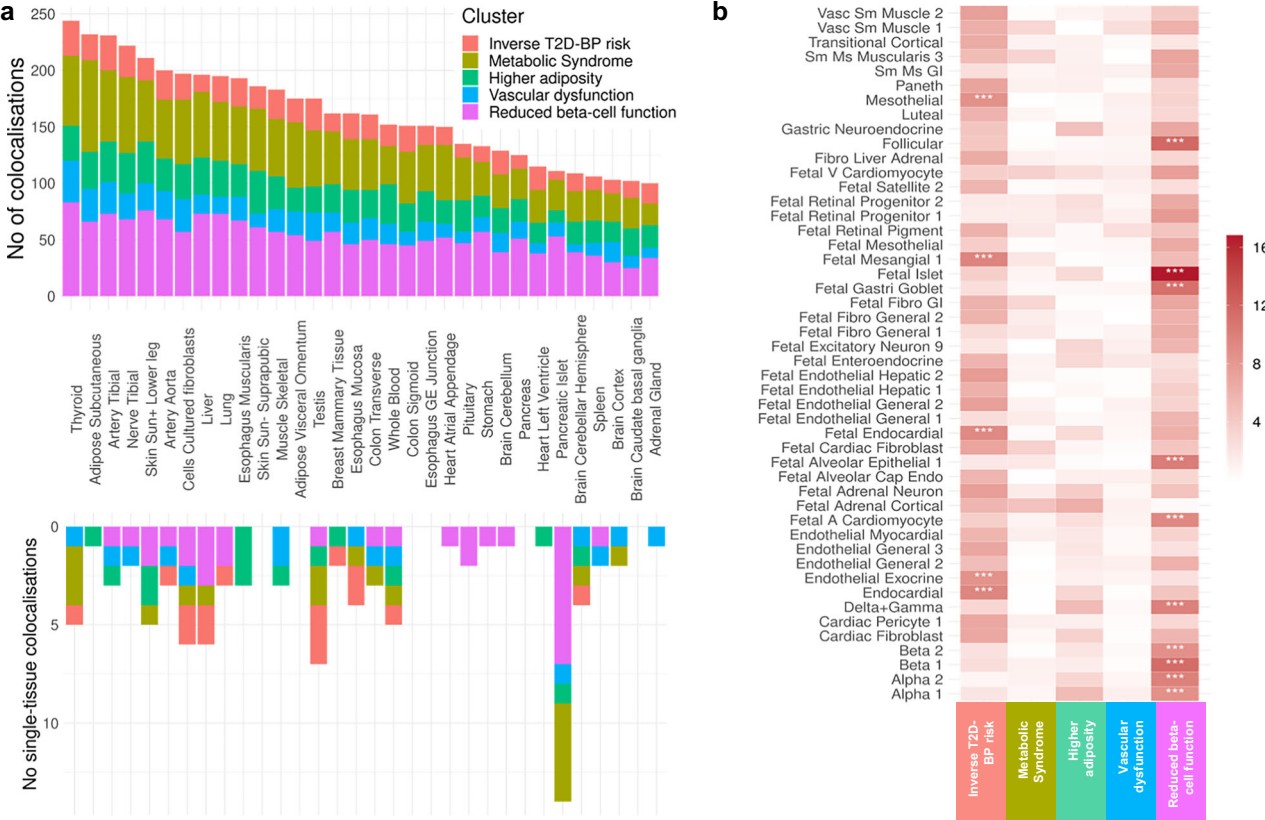

**Fig. 3 | Characterisation of pathogenetic clusters: genetic expression, and regulatory mechanisms. a** Histogram depicting the distribution of colocalized loci across clusters in 50 human adult tissues (Supplementary Data 5, 6). Each bar represents the number of Bayesian colocalized signal (two-sided; PP.H4 > 0.8 and PP.H3 < 0.5) per tissue, with colours indicating contribution from the five clusters. The upper histogram shows the overall number of colocalizations while the lower portion highlights tissue-specific colocalizations. **b** Heat map showing the enrichment of open chromatin region across the five clusters and 222 cell types from 30 human adult tissues and 15 human fetal tissues (Supplementary Data 7). Each column corresponds to a cluster, and each row to a cell type. Colour intensity represents the $-\log(p$ value) from two-sided Chi-squared tests comparing nested logistic regression models assessing enrichment. Asterisks denote significant signals following Bonferroni correction for multiple testing ($p < 2.25 \times 10^{-4}$). No number, T2D type 2 diabetes, BP blood pressure, Sun+ sun-exposed, Vasc vascular, Sm smooth, Ms muscle, GI gastrointestinal, V ventricular, Fibro fibroblast, A atrial.

Our analysis revealed distinct gene expression signatures for each cluster, corroborating the diversity of the biological pathways involved. The *Inverse T2D-BP risk* cluster displays colocalization in the brain, particularly brain cerebellum (*MGRN1*, *HELLS*, *SLC39A13*) and adrenal glands (*SLC7A1*, *RHOC*, *NUDT2*). The *Metabolic Syndrome* cluster variants colocalized in adipose subcutaneous (*JAZF1*, *ALKAL2*, *LCORL*). The *Higher adiposity* cluster shows colocalization in skin (*MYO19*, *EIF3C*, *SLC39A10*) and whole blood (*WFS1*, *CCDC134*, *MED27*). The *Vascular dysfunction* cluster colocalized with fibroblasts (*ERI1*, *FOXD4*, *RSRC1*) and thyroid (*CSTB*, *ZNF638*, *SNX31*) and the *Reduced beta-cell function* cluster with pancreatic islets (*C2CD4B*, *ADCY5*, *PHB*).

We identified 99 tissue-specific colocalizations (Fig. 3a bottom, "Methods"). While thyroid and adipose subcutaneous tissues showed a high number of total colocalizations, pancreatic islets showed the highest number (14) of single-tissue (i.e., specific) colocalizations, particularly among the clusters strongly associated with risk of T2D such as the *Reduced beta-cell function*, *Metabolic Syndrome* and *Higher adiposity* clusters. Notably, the pancreatic islet tissue-specific colocalized genes include *TH* (synthesis of catecholamines)[48] in the *Higher adiposity* cluster, *MTNR1B* (circadian rhythms and glucose metabolism)[49], *FXYD2* (Na,K-ATPase pump regulator)[50], *G3BP2* (cellular stress)[51] in the *Reduced beta-cell function* cluster, *SYNDIG1L* (synapse development), *LTBP3* (cell growth, differentiation and repair)[52], *CLEC18A* (immune function)[53] in the *Metabolic Syndrome* cluster

(Supplementary Data 6). This demonstrates the predominant role of pancreatic islets in T2D pathogenesis and its related complications.

To explore the underlying mechanisms in the *Inverse T2D-BP risk* cluster, we conducted pathway analysis using *Metascape*[54] for the 202 colocalized genes identified within this cluster (Supplementary Fig. 8). This analysis revealed an overwhelming enrichment in the retinol metabolic process (GO:0042572), which involves one of three compounds that make up vitamin A (retinol, retinal, and retinoic acid). All components of the retinol metabolism are associated with both T2D and CVD[55].

We then dissected the localisation of our SNVs in a cluster-specific manner using chromatin accessibility atlases from CATLAS, based on scATAC-seq peaks. The atlas encompasses 222 cell types from 30 human adult tissues and 15 fetal tissues, allowing examination of the enrichment of candidate cis-regulatory elements (cCREs) in each cluster across different cell types (Fig. 3b and Supplementary Data 7). The clusters were enriched in diverse regulatory mechanisms. Specifically, the *Inverse T2D-BP* cluster exhibited significant ($p \leq 2.25 \times 10^{-4}$) enrichment for regions of open chromatin in mesothelial cells, endocardial cells, endothelial in exocrine tissue cells as well as fetal endocardial and mesangial cells. The *Reduced beta-cell function* cluster demonstrated strong enrichment in several fetal cell types, such as islets, gastric goblet, alveolar epithelial, cardiomyocyte, as well as follicular cells and cells from the pancreas tissues, including delta,

gamma, beta and alpha. This suggests that, beyond islet dysregulation and insulin impairment, pathways involved in fetal development also play an important role in adult metabolic health. Moreover, nominal enrichments were observed in fetal adrenal cortical cells for the *Metabolic Syndrome* and *Higher adiposity* clusters, suggesting hormone regulatory implications beginning as early as intrauterine development[56].

Given the large number of colocalized SNVs observed across clusters in tissues such as the thyroid, subcutaneous adipose tissue, tibial artery, tibial nerve, and lower leg skin (Fig. 3a), we further explored the colocalized genes by identifying their enrichment in the primary cell types corresponding to these tissues—namely, follicular cells, adipocytes, smooth muscle cells, Schwann cells, keratinocytes (Fig. 3b). Notably, we identified 15 genes that both colocalized in thyroid and were enriched in follicular cell cCREs (Supplementary Table 3). While some of these genes were previously associated with T2D such as *CAMK1D* (energy homoeostasis and beta-cell receptor signalling pathway)[57] or *KCNH6* (insulin secretion and glucose homoeostasis)[58], and with BP regulation such as *ACE* (renin-angiotensin system)[28], the remaining are potential candidate genes for the T2D-BP pathogenesis. Among them are *SAE1* (known in cancer)[59], *GSAP* (known in Alzheimer's disease)[60], *DCAF7* (cellular differentiation)[61], *MAP3K3* (stress and inflammation)[62]. Subsequent *Metascape*[54] pathway analysis highlighted fundamental cellular processes, including protein ubiquitination (GO:0016567) and positive regulation of protein modification process (GO:0031401). The overlap of signals between colocalized genes and cCREs in the other four tissues consistently highlighted the *MAP3K3* gene. Subsequent pathway analyses did not yield conclusive results (Supplementary Table 4).

Overall, changes in gene expression within clusters highlighted the importance of the thyroid tissue in T2D-BP shared pathophysiology and retinol metabolism within the *Inverse T2D-BP risk* cluster, while the high number of fetal cell regulatory elements enrichment suggests a strong contribution of intrauterine growth pathways in both T2D and high BP.

## T2D-BP comorbidity using partitioned PGSs

To evaluate the ability of the T2D-BP SNV clusters to predict comorbidities and complications, we used the individual-level data from UKB and built unweighted partitioned PGSs for each cluster group. Using the R software environment tool, *comorbidPGS*, we aligned partitioned PGS to the T2D risk allele, i.e., each allele increasing the risk of T2D is counted as one in the PGS calculation ("Methods")[24].

To illustrate the influence of genetic predisposition, we computed the relative risk of comorbidity among the UKB individuals ("Methods"), based on the top 10% percentiles of each unweighted partitioned PGS. Whereas the overall prevalence of T2D-high BP comorbidity in the UKB was 5.49% (Fig. 4a), individuals in top 10% of the unweighted risk score of *Higher adiposity*, *Metabolic Syndrome*, and *Reduced beta-cell function* clusters had a relative risk RR[95% CI] of 1.36[1.32–1.41], 1.44[1.39–1.48], 1.55[1.51–1.60], respectively. Moreover, the individuals in the top 10% distribution of *Metabolic Syndrome* and *Reduced beta-cells function* combined PGSs, derived from 536 SNVs, showed a 2.13[1.96–2.31] fold increased risk of having comorbidity (Fig. 4b and Supplementary Table 5), reaching the same RR as a traditional pruning-and-thresholding (P + T) weighted T2D PGS. Survival analysis, using cumulative hazard plots, indicated that this elevated comorbidity risk was consistent and linear over 15 years of follow-up (year 0 representing the date of first diagnosis with either hypertension or T2D, Fig. 4c and Supplementary Fig. 11). This suggests that individuals with high PGS distributions remain at increased risk of comorbidity throughout their life course. Consequently, partitioned PGSs enhance the predictive ability to identify high-risk individuals at an earlier age[63,64].

Using the partitioned PGSs, we evaluated the association between PGS and multiple sets of complications based on the UKB hospital records (Fig. 5 and Supplementary Data 8). We detected a reciprocal protective effect of the *Inverse T2D-BP* cluster PGS on essential hypertension (OR[95% CI] = 0.91[0.90–0.92], $p < 1.00 \times 10^{-40}$) alongside other circulatory system disorders such as coronary artery disease (CAD, OR[95% CI] = 0.94[0.93–0.95], $p = 5.08 \times 10^{-22}$), angina pectoris (OR[95% CI] = 0.95[0.94–0.97], $p = 1.29 \times 10^{-12}$), AF (OR[95% CI] = 0.96[0.94–0.97], $p = 1.01 \times 10^{-10}$), chronic ischaemic heart disease (OR[95% CI] = 0.96[0.95–0.97], $p = 9.08 \times 10^{-10}$). The *Inverse T2D-BP risk* cluster PGS also associates with lower risk of gout (OR[95% CI] = 0.92[0.89–0.95], $p = 1.10 \times 10^{-6}$) and hypercholesterolaemia (OR[95% CI] = 0.98[0.97–0.99], $p = 1.90 \times 10^{-17}$). These results support previous research on the heterogeneous effects of hypertensive medications on the risk of T2D, indicating that some biological processes between T2D and high BP may reduce the risk of comorbidity[16].

We confirmed the high contribution into T2D-BP comorbidity of the *Metabolic Syndrome* cluster, by showing that its relatively small number of SNVs could predict risk of multiple metabolic disorders, including T2D (OR[95% CI] = 1.24[1.23–1.26], $p < 1.00 \times 10^{-40}$), hypertension (OR[95% CI] = 1.13[1.12–1.14], $p \leq 1.00 \times 10^{-40}$), hypercholesterolaemia (OR[95% CI] = 1.10[1.09–1.10], $p \leq 1.00 \times 10^{-40}$), hyperlipidaemia (OR[95% CI] = 1.10[1.08–1.13], $p = 6.52 \times 10^{-20}$), fatty liver (OR[95% CI] = 1.13[1.09–1.18], $p = 6.08 \times 10^{-10}$), and hypothyroidism (OR[95% CI] = 1.03[1.02–1.04], $p = 1.65 \times 10^{-6}$). Additionally, the *Metabolic Syndrome* cluster PGS showed significant association with risk of cardiovascular complications such as CAD, angina pectoris, ischaemic heart disease, heart failure, and myocardial infarction. We also detected association with higher risk of kidney failure and calculus of kidney.

The *Higher adiposity* PGS showed the strongest risk prediction of obesity-related diseases, including T2D (OR[95% CI] = 1.22[1.20–1.23], $p \leq 1.00 \times 10^{-40}$), hypertension (OR[95% CI] = 1.09[1.09–1.10], $p \leq 1.00 \times 10^{-40}$), sleep apnoea (OR[95% CI] = 1.17[1.14–1.20], $p = 1.28 \times 10^{-34}$), osteoarthritis (OR[95% CI] = 1.08[1.07–1.10], $p = 1.51 \times 10^{-31}$), carpal tunnel syndrome (OR[95% CI] = 1.09[1.07–1.11], $p = 6.61 \times 10^{-20}$), and pneumonia (OR[95% CI] = 1.08[1.06–1.10], $p = 3.92 \times 10^{-12}$). This cluster PGS was associated with higher mental disorders, such as major depressive disorder, delirium or behavioural disorders due to use of tobacco, highlighting the intertwined relations between obesity and depressive conditions.

Among other clusters, the *Vascular Dysfunction* cluster was more predictive for cardiovascular complications, including hypertension (OR[95% CI] = 1.12[1.11–1.13], $p < 1.00 \times 10^{-40}$), CAD (OR[95% CI] = 1.06[1.05–1.07], $p = 4.25 \times 10^{-23}$), or ischaemic heart disease (OR[95% CI] = 1.06[1.04–1.07], $p = 1.05 \times 10^{-15}$). The *Reduced beta-cell function* unweighted PGS showed the strongest association with risk of T2D (OR[95% CI] = 1.33[1.32–1.35], $p < 1.00 \times 10^{-40}$) and its related consequences, including obesity (OR[95% CI] = 1.03[1.02–1.05], $p = 4.38 \times 10^{-9}$), hyperlipidaemia (OR[95% CI] = 1.08[1.06–1.10], $p = 5.17 \times 10^{-13}$), fatty liver (OR[95% CI] = 1.09[1.05–1.14], $p = 1.62 \times 10^{-5}$), chronic kidney disease (OR[95% CI] = 1.10[1.07–1.12], $p = 6.47 \times 10^{-19}$), and hypothyroidism (OR[95% CI] = 1.03[1.01–1.04], $p = 2.43 \times 10^{-5}$). The *Reduced beta-cell function* weighted PGS ("Methods") showed strong association with high SBP and PP, albeit not with DBP (Supplementary Data 9).

The partitioned PGSs effectively delineated the differences in prediction among the T2D-BP cluster SNVs. Partitioned PGSs shows that grouping of SNVs can highlight related comorbidities through different pathophysiological processes.

## Discussion

In this large-scale multiomic study, we explored the complex genetic underpinnings of the comorbid relationship between T2D and high BP. Our analysis confirms and extends prior evidence of a direct genetic

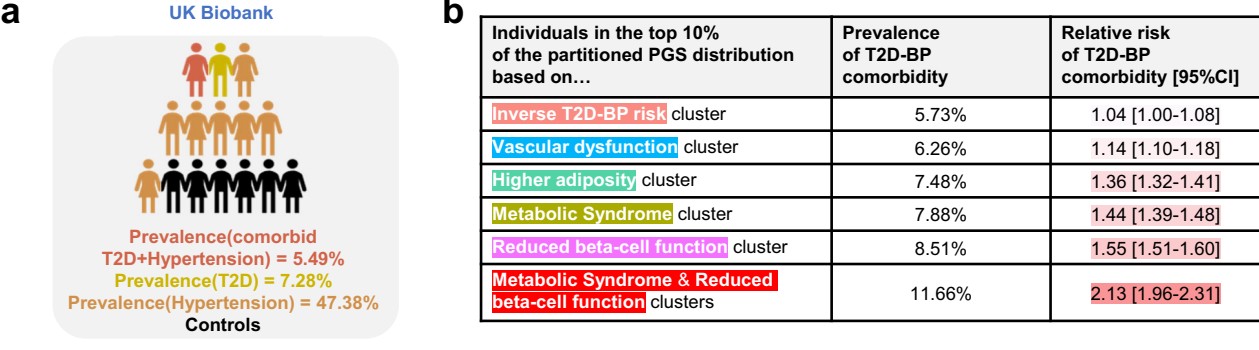

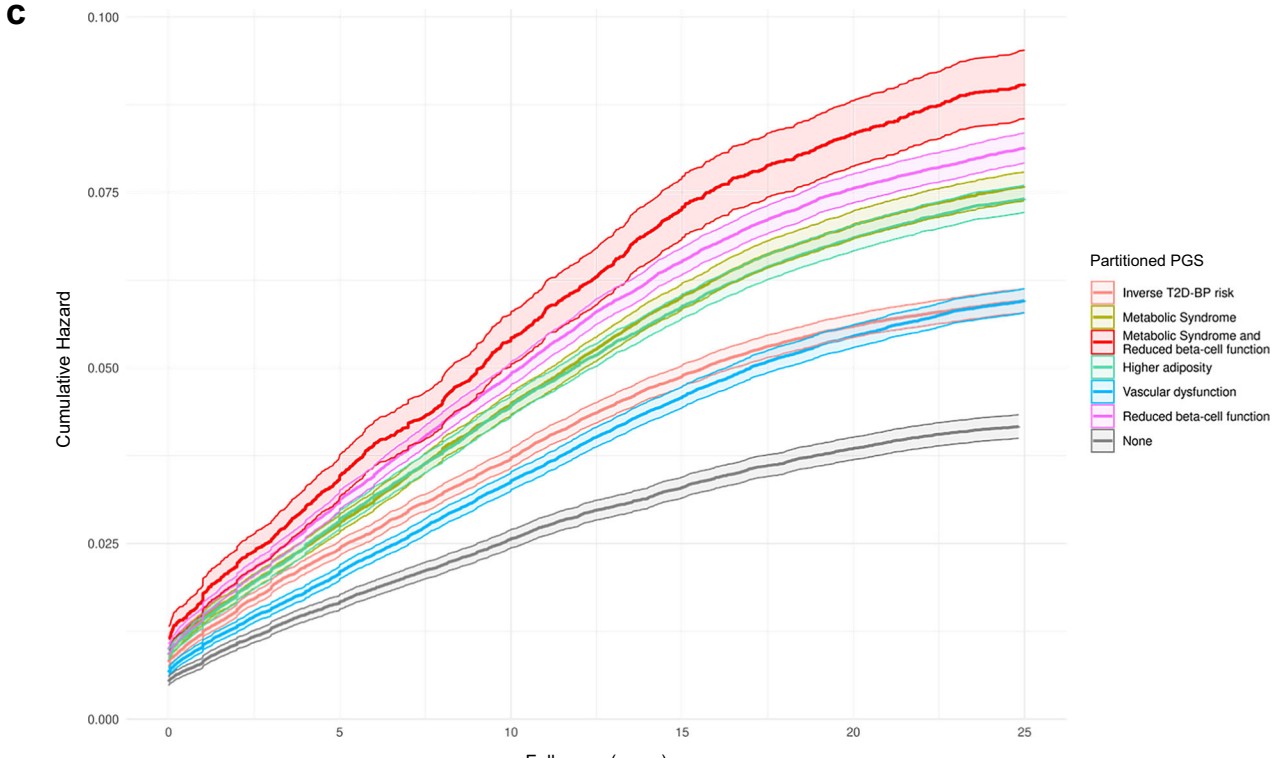

**Fig. 4 | The T2D-BP comorbidity risks stratified by partitioned PGSs in the UKB.**
**a** Overall prevalence of individuals with T2D, hypertension, and T2D-BP comorbidity (Supplementary Table 5). Prevalence (Prev = $\frac{N_{cases}}{N_{total}}$) corresponds to the proportion of cases divided by the total number of individuals. **b** Summary table of the relative risk (RR = $\frac{Prev_{cases, cluster}}{Prev_{cases, overall}}$) of T2D-BP comorbidity for individuals in the top 10% of cluster-specific PGSs in the UKB, with 95% confidence intervals (CIs) derived using the Wald method on the log-transformed RR (Supplementary Table 5). Two-sided tests were applied throughout. **c** Cumulative hazard of T2D–BP comorbidity stratified by being in the top 33% of the unweighted partitioned PGS after clustering (Supplementary Data 10). Shaded areas represent the 95% CI of the two-sided estimated cumulative hazard from the Cox proportional hazards model. Prev prevalence, comorbid T2D-BP comorbidity, T2D type 2 diabetes, BP blood pressure, CI confidence interval.

correlation and a large overlap in genetic signals shared between T2D and high BP[65,66]. This observation is not unexpected, given the well-established comorbidity between T2D and hypertension—largely attributable to shared environmental and biological risk factors such as adiposity. Our approach extends beyond this by leveraging clustering of genetic variants to partition the T2D-BP genetic architecture into biologically coherent groups.

We curated a set of genome-wide significant common SNVs associated with T2D and high BP, thereby enriching for variants with stronger and trait-specific effects, and reducing the influence of broader, less specific cross-trait associations[67,68]. Through hierarchical clustering using T2D-BP related endpoints and risk factors, we identified five clusters of SNVs, each highlighting unique pathogenetic processes underlying the T2D-BP relationship. This clustering approach provided a clearer delineation of genetic relationships, reducing heterogeneity compared to other clustering methods. Four of these SNV clusters—*Metabolic Syndrome*, *Higher adiposity*, *Vascular dysfunction*, and *Reduced beta-cell function*—align with established findings in high BP or T2D[27,69–71]. We discovered an intriguing cluster of variants with an *Inverse T2D-BP risk* profile, implicating retinol metabolism. Although recent meta-analysis on the role of retinol in T2D and BP regulation have yielded inconsistent results, one retinol derivative, retinoic acid, has consistently been linked to higher IR and reduced cardiovascular events[55]. While we propose a mechanistically plausible pathophysiological hypothesis for inverse T2D-BP risk effects, these patterns may also arise from incomplete overlap in the genetic architectures underlying T2D and BP. The *Metabolic Syndrome* cluster was associated with features such as shorter stature, higher WHR, and no detectable effect on BMI. These features are consistent with the International Diabetes Federation (IDF) definition of MetS[72], which emphasises central obesity—better captured by WHR than

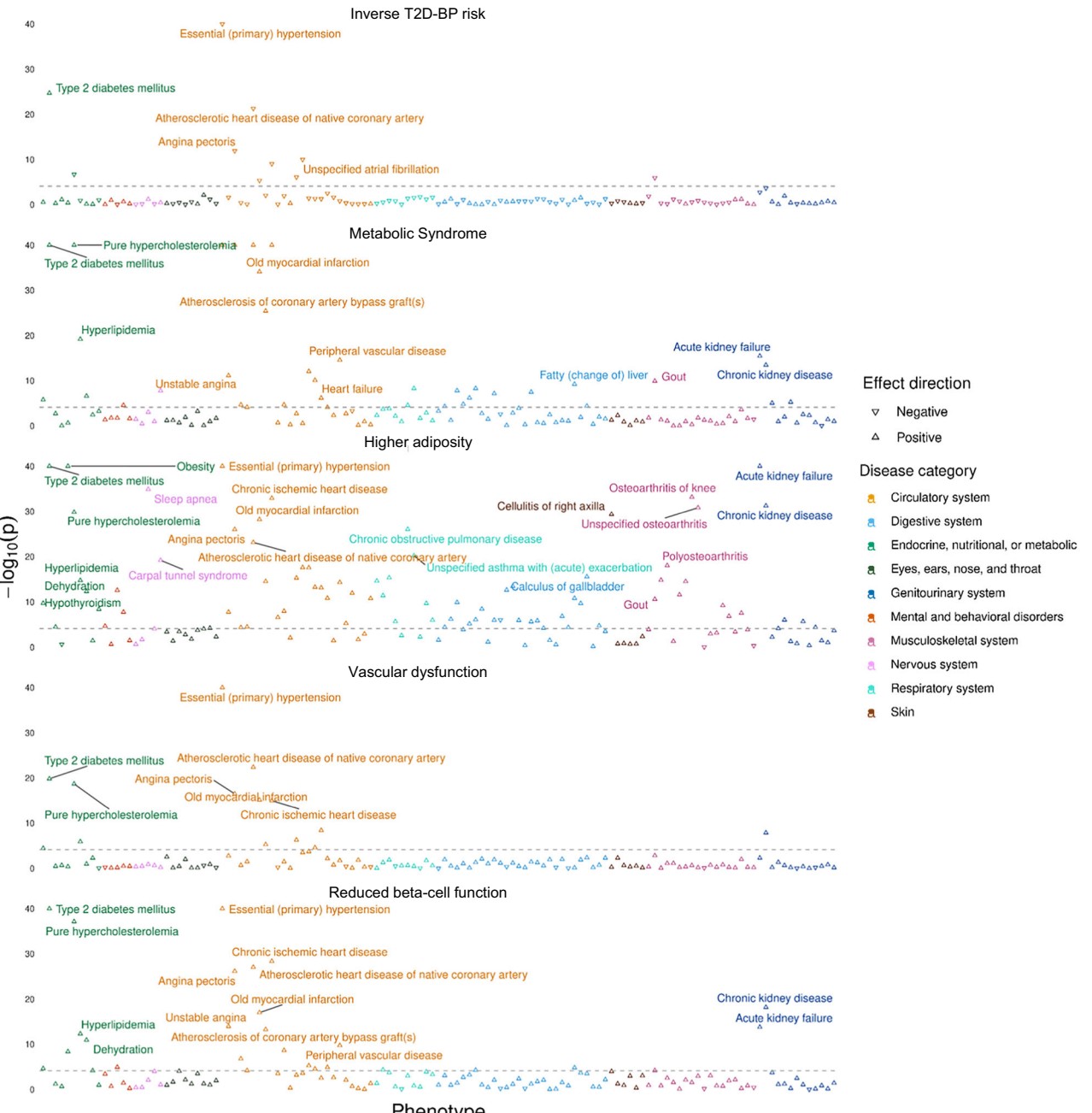

**Fig. 5 | Association between complications and partitioned PGSs after clustering in the UKB.** The figure displays five Manhattan plots, each representing results from two-sided logistic regression analyses testing the association between cluster-derived partitioned PGSs and disease risk (Supplementary Data 8). The grey dotted line indicates the threshold for significant associations applying Bonferroni multiple testing correction ($p < 7.75 \times 10^{-5}$). Arrows illustrate the direction of association between PGS and disease risk: upward-pointing arrows indicate that a higher PGS is associated with an increased risk of the disease, while downward-pointing arrows indicates that a higher PGS is associated with a decreased risk of the disease. Colours differentiate the different disease categories. PGS polygenic score, UKB UK Biobank, T2D type 2 diabetes, BP blood pressure.

BMI—as the primary driver. No detectable effect on BMI in this cluster likely reflects the specific contribution of visceral adiposity to cardiometabolic risk, rather than overall body adiposity, highlighting the relevance of WHR-linked pathways in MetS pathophysiology. Additionally, we observed an enrichment in colocalizations specific to the thyroid across all five T2D-BP clusters, suggesting a mechanistic role for thyroid function in T2D-BP comorbidity. This feature further highlights a need for better thyroid health in the general population to reduce impact of dysthyroidism on BP and T2D management[73].

We bring forward a property of partitioned PGS to differentially predict related T2D–high BP conditions[74]. While PGS has shown predictive value for an expanding array of common diseases, such as for instance CAD[75,76], its clinical application remains limited due to a range of pitfalls[77]. In this study, we report how specific clusters, particularly the *Metabolic Syndrome* and *Reduced beta-cell function*, can identify a sub-population of individuals bearing over twice the general population risk of T2D-BP comorbidity. Our study calls for partitioning of PGSs to predict complications in metabolic disorders like T2D and high

BP, suggesting that in comorbidity risks, fewer SNVs could be more impactful than an entire genome-wide PGS to stratify the individuals at high risk of complications. For the survival analysis, individuals were assigned to the cluster corresponding to their highest partitioned PGS, to emulate a potential clinical framework where patients could be stratified into their predominant mechanistic risk pathway for targeted prevention. While this approach simplifies the genetic architecture, it illustrates how refined, cluster-specific PGSs could ultimately complement existing clinical tools such as QDiabetes[78] and QRisk[79] to enable earlier, pathway-informed interventions. This work paves the way for improved risk stratification and precision health approaches[76].

We acknowledge several caveats in this work. The pathophysiological mechanisms involved in both T2D and high BP are not fully explained by genetics alone, and are influenced by a variety of external and environmental factors, such as salt consumption or western diet. Moreover, the heterogeneity in sample origins, study designs, and sample sizes across the diverse included datasets could potentially reduce our ability to identify independent pathways. While the inclusion of individuals with potentially pre-existing comorbid conditions in GWAS studies of specific phenotypes, here T2D and BP, is often inevitable to capture the genetic architecture, it could somewhat inflate the estimated shared genetics due to phenotypic correlation. To mitigate the possibility of sample contamination bias, we used different subsets of T2D-BP GWASs for analyses that are sensitive to linkage disequilibrium structure. We mitigated the sample origin bias by prioritising datasets with multiple ancestries including European[80]. Hierarchical clustering can force an SNV to fit a cluster that may not fully capture its effect. To mitigate this, we compared and found consistent results with previous clustering studies and sensitivity analyses using other clustering methods, such as bNMF[27]. Finally, we chose to focus on clustered genome-wide significant common variants to provide a clear and interpretable framework for dissecting the T2D-BP shared genetic relationships. While this strategy highlights robust mechanistic clusters, it may overlook a sizeable portion of the polygenic signal. Future studies could apply complementary approaches such as expanding this study to whole-exome or whole-genome datasets, or using multivariate GWAS or *genomicSEM*[40,81] to further refine the understanding of T2D-BP relationship.

In this study, we highlight the mechanistic heterogeneity underlying T2D and high BP, demonstrating that their genetic relationship is not driven by a single shared pathophysiological process but instead arises from five distinct mechanistic pathways contributing to the T2D-BP comorbidity. The partitioned PGSs, derived through clustering, enable modelling of differential lifetime risk trajectories associated with T2D-BP comorbidity. These results provide a framework for future investigations into stratified risk prediction and pathophysiology-informed approaches to manage comorbid conditions.

## Methods

### Material description
We collected 45 publicly available GWAS summary statistics (Supplementary Data 2), spanning from 2017 to 09 June 2023. GWAS were selected if they included a large number of participants ($N \geq 10,000$) and a preference for datasets with diverse ancestral backgrounds, involving the majority of individuals of European ancestry, to enhance genetic diversity in our study. European-only studies were included when multi-ancestry data were unavailable. Particularly for T2D, we used Mahajan et al.[22] GWAS (2018b) as it allows us to alternate between a GWAS with UKB individuals or not, accessible on the DIAGRAM/DIAMANTE/T2DGGI webpage https://diagram-consortium.org/downloads.html. We alternated between using the Warren et al. GWAS[71] based on the UKB and a subset version of the Evangelou et al. GWAS[12] without UKB (only ICBP data). Both T2D/BP GWASs without

UKB individuals were used only for two specific analyses: LD score regression and PGS.

To ensure robustness in uncovering shared aetiologies, we gathered from multiple sources a list of T2D[82–86] and BP SNVs[13,30,87–97] reaching genome-wide significance (i.e., $p < 5 \times 10^{-8}$) and characterised by minor allele frequency (MAF) > 0.01. SNV could have been reported for either one of the three BP metrics (SBP/DBP/PP) or it was reported for T2D. We identified a total of 1401 SNVs. This list was further curated for clustering to keep only independent SNVs (LD $r^2 < 0.2$). We did not apply additional pruning based on genetic distance. This decision was made to preserve the potential for capturing pleiotropic variants and regional architectures that could contribute to comorbidity between T2D and BP traits. The resulting list encompasses 1304 SNVs, including 500 T2D and 283/272/270 SBP/DBP/PP signals, with 9 unique SNVs demonstrating associations with T2D-BP comorbidity.

The UK Biobank (UKB, https://ukbiobank.ac.uk/) is a large prospective cohort study with genotypic and phenotypic data[21]. The dataset in this study derived from genome-wide imputed data, including 459,247 individuals of European ancestry. To identify individuals with T2D, we leveraged hospital admission records, self-reports, and ICD10/9 codes, successfully defining 33,446 T2D individuals. Hypertension was assessed using three blood pressure metrics: SBP, DBP, and PP, with both automated and manual records available in the UKB[12]. We used the mean value when more than one value was found for a given individual and adjusted for medication use by adding 15 mmHg to SBP and 10 mmHg to DBP for individuals with reported data on blood pressure-lowering medication[98]. We defined the outcome "hypertension" used in the last section of the *Results* (unweighted PGS evaluation) as either having SBP ≥ 150 mmHg, or DBP ≥ 90 mmHg, or taking blood pressure lowering medication. The complete criteria used to identify individuals with T2D and hypertension are illustrated in Supplementary Fig. 10. We identified 217,599 hypertensive individuals in the UKB (Supplementary Table 1).

The Genotype-Tissue Expression (GTEx, https://www.gtexportal.org/home/datasets/) is a project gathering samples from 49 non-diseased tissue across 1000 deceased individuals[17]. We used the expression quantitative trait *loci* (eQTLs) mapping genetic variants with changes in the expression of nearby genes.

The ABOS cohort is an ongoing prospective study that aims to identify the determinants of bariatric surgery outcomes, initiated at Lille University Hospital (Lille, France) in 2006. The study protocol has been previously detailed elsewhere (clinicaltrials.gov, NCT01129297)[19]. A total of 372 individuals of European descent were included in the ABOS liver eQTL study.

The translational human pancreatic islet genotype tissue-expression resource (TIGER, http://tiger.bsc.es/) is a large meta-analysis of cohorts aggregating more than 500 human islet genomic datasets from five cohorts in the Horizon 2020 consortium T2DSystems[18].

The cis-element ATLAS (CATLAS, http://catlas.org/humanenhancer/) is a comprehensive resource gathering the genome-wide cCREs in 222 human cell types. The dataset encompasses chromatin accessibility data derived from single-cell Assay for Transposase-Accessible Chromatin (ATAC-seq) peaks spanning 30 human adult and 15 human fetal tissues.

We systematically applied multiple correction testing and adjusted the appropriate individual-level associations for age, sex, genotyping array and the first six principal components derived from genetic data.

### Genetic correlation
We conducted genetic correlation analysis by using the whole T2D[22] and BP[12] GWAS datasets of European ancestry only without UKB individuals. We estimated the liability-scale heritability ($h_2$) of each

phenotype and their genetic correlation ($r_g$) by employing linkage disequilibrium score regression via *ldsc* v1.0.1[23]. We used pre-computed LD scores from 1000 Genome Phase 3 SNVs in individuals of European ancestry. We manually aligned the GWAS datasets and excluded outlier SNVs, that are multi-allelic, poorly-imputed (by filtering to HapMap3 SNVs), have a MAF > 0.01 and $0 < p \leq 1$.

## Reciprocal risk prediction using PGS

We constructed PGS before and after clustering with *plink* v1.9[99], using specific SNV sets, weights from T2D[22], SBP, DBP, and PP[12] summary statistics (GWAS without UKB individuals) and individual-level data from the UKB. We carefully selected weights without UKB (Supplementary Data 2 in italic) to avoid sample overlap between the base data (composed of GWAS significant SNVs plus their weights) and the target data (individual-level data from the UKB). The Pruning and Thresholding (P + T) method was used before making PGS by selecting the only GWAS significant SNVs for each desired outcome[100,101]. We subsequently used *comorbidPGS* v1 to conduct linear or logistic regression to evaluate the shared predisposition between PGS for the *i*-th trait and the *j*-th target phenotype *Y*, correcting for covariates including age, genetically-inferred sex, genetic array, and the first six principal components as following[24]:

$$E\left(Y_{kj}\right) = \alpha_i PGS_{ki} + \alpha_{age} age_k + \alpha_{sex} sex_k + \alpha_{array} array_k$$
$$+ \sum_{x=1}^{6} \alpha_x PC_{kx} + \alpha_0$$

Where $Y_{kj}$ is the value of the *j*-th phenotype of the *k*-th individual (1 or 0 for T2D, continuous value in mmHg for BP traits), $PGS_{ki}$ is the PGS for the *i*-th trait and the *k*-th individual, $age_k$, $sex_k$, $array_k$ and $PC_{kx}$ being the respective covariates for the *k*-th individual. The $\alpha$ are the regression coefficients, $\alpha_0$ is the intercept.

## Genetic overlap

We estimated the genetic overlap among the pre-curated non-independent 1401 SNVs by identifying variants within 500 kb of each other, or in LD $r^2 > 0.2$ for individuals of European ancestry. The reported *loci* are the proximal ones, identified by *biomaRt*[102]. We identified and coloured the signals per clusters in Supplementary Data 1.

## Clusters of pathogenetic processes

We aggregated the 1304 independent T2D-BP SNVs and investigated 45 endophenotypes GWAS summary statistics including T2D/BP (Supplementary Data 2) to cluster them into distinct groups based on pathogenetic processes. To reduce/avoid missingness across datasets, we identified high-LD proxies (LD $r^2 > 0.6$) using European ancestry reference panel from the 1000 Genomes Project Phase 3. Although some of the GWAS datasets were of multiple ancestries, we prioritised datasets comprising European-only studies or those with a high proportion of European ancestry participants. This approach mitigated potential discrepancies in LD patterns between the GWAS datasets and the reference panel. For the *i*-th SNV of the *j*-th trait, we extracted the beta coefficient, $\beta_{ij}$, along with its standard error, $s_{ij}$, to build z-scores using the following formula $Z_{ij} = \frac{\beta_{ij}}{s_{ij}}$. Any signal exhibiting more than 20% missingness across the GWAS was excluded. We conducted imputation of the remaining missing z-scores using a random forest algorithm from *imputeSCOPA*[34]. For each GWAS, we truncated SNVs if their absolute z-score exceeded two standard deviations.

We performed hierarchical clustering with the Ward method from the R function *hclust*, using Euclidean distance and the R package *pheatmap* for plotting (Supplementary Fig. 1). We determined the number of clusters by taking half of the maximum Euclidean tree-row distance. Our methodology, termed 'hard clustering', ensures precise assignment of each SNV to a single cluster. Each variant is treated as a

unit, in opposition with approaches that treat positive and negative associations independently. Using a 'hard clustering' method offers a clearer delineation of clustering patterns. However, it may force a SNV into a group where is does not truly belong[9,37].

To validate our clusters, we compared them with those identified in the latest T2DGGI study[9] and both latest 'soft' clustering for T2D[37] and BP[38]. First, we looked at SNV assignment between our T2D-BP clusters, T2DGGI clusters (Supplementary Fig. 5a), latest T2D 'soft' clusters (Supplementary Fig. 5b), and latest BP 'soft' clusters (Supplementary Fig. 6 and Supplementary Data 3). To do so, we systematically looked for LD proxies (LD $r^2 > 0.6$) in the three clusters. We assign a SNV into 'soft' clusters based on weight >0.75 (Supplementary Data 3 and Supplementary Figs. 5, 6). Second, we performed GWAS weight comparison specifically between our T2D-BP hierarchical clusters ($k = 5$) and the T2DGGI hierarchical clusters ($k = 8$) by extracting each cluster weights for the common GWAS endophenotypes. We then calculated the Pearson correlation coefficient between the sets of trait cluster weights. Notably, our T2D-specific clusters showed significant correlations: the *Higher adiposity* cluster aligned with T2DGGI 'Obesity' cluster, the *Metabolic Syndrome* cluster was correlated with 'Lipodystrophy', 'Metabolic syndrome', and 'Residual glycaemic'. The *Reduced beta-cell* cluster exhibited correlation across all T2DGGI clusters, particularly with 'Residual glycaemic' and 'Obesity' (Supplementary Fig. 7).

Additional sensitivity analyses include the use of three alternative clustering methods: adjusting Z-score for GWAS sample size prior to clustering (using the following formula $Z_{ij} = \frac{\beta_{ij}}{s_{ij} \times \sqrt{N_j}}$, Supplementary Fig. 2), *MRClust*[36] (Supplementary Fig. 3) and a 'soft clustering' method, bNMF[27] (Supplementary Fig. 4). In Supplementary Fig. 2a, b, we showed a comparative hierarchical clustering using our approach without GWAS sample size adjustment, and another approach by adjusting for sample size (dividing $Z$ by $\sqrt{N}$, $N$ being the sample size for a given GWAS), as it has been done in other studies[9]. We decided to use unadjusted Z-scores as main analysis as we wanted to prioritise robust, high-confidence associations. We showed in Supplementary Fig. 2c that the cluster assignment has followed the same patterns with or without adjustment (Chi-squared test of independence $p < 2 \times 10^{-16}$). We compared *our bNMF* results ($k = 8$) with our hierarchical clustering ($k = 5$) by merging the variant weights for each cluster of both methods. Logistic regression models were then employed to assess the association between the 'hard' cluster memberships and the 'soft' clustering weights. Specifically, for each hierarchical cluster, we modelled the binary cluster membership (0/1) as a function of the corresponding bNMF-derived weights. The models' deviances were compared through ANOVA to evaluate the enrichment and significance of the clustering concordance (Supplementary Fig. 4c).

To improve interpretability of the clustering, we performed linear regression with the *lm* R function, across the SNVs within the *k*-th cluster on the *i*-th phenotype, defined by $E\left(Z_{ij}\right) = \sum_k \alpha_{ik} C_{kj}$, where $C_{kj}$ is a variable taking the value 1 if the *j*-th SNV was assigned to the *k*-th cluster, 0 otherwise. The resulting regression (Supplementary Data 4) is represented in Fig. 2 heatmap, with the *p* value denoting the colour intensity, and the colour indicating the sign of the regression coefficient. PAI 1, oestradiol (f and m), and age at menopause are not shown in Fig. 2 to improve readability.

## Colocalization analysis

To identify the signals that share the same causal variants in T2D-BP GWASs and eQTLs, we systematically conducted Bayesian colocalization with *coloc.abf*[47], between the clustered SNVs and eQTLs derived from 50 tissues from GTEx (48 tissues), ABOS (liver) and TIGER (pancreatic islet) datasets. We used an hypothesis-free approach. We assessed colocalization with the 'lead' GWAS per cluster (i.e., the main origin of the clustered SNVs, SBP GWAS for *Inverse T2D-BP* and *Vascular dysfunction*, T2D GWAS for *Metabolic syndrome, Higher adiposity*

and *Beta cell* clusters). For each SNV within the 1304 clustered SNVs, we considered a genomic region of 100 kb up and downstream, and evaluated whether the variant colocalized with genetic expression in that region. Colocalization was asserted if the region was well-characterised (containing between 100 and 1000 SNV within eQTL), the posterior probability of the model sharing a single causal variant (H4) exceeded 80%, and the posterior probability of distinct causal variants with both traits (H3) was below 50%. We excluded variants within the MHC region (chr6: 25,000,000–35,000,000).

We present in Fig. 3a the count of colocalized genes across tissues (bar) and clusters (colours). We represented in this figure the top tissues based on their overall number of colocalization ($N_{coloc} \geq 100$). The lower portion depicts the number of tissue-specific colocalizations, i.e., when the index SNV is associated to only one tissue-specific eQTL. The complete number of colocalization per cluster and tissue is represented in Supplementary Fig. 8. The colour intensity indicates the percentage of colocalized genes per cluster. Caution is warranted in interpreting the results as there is currently no robust method to rule out horizontal pleiotropy[103].

Subsequent pathway analysis of the thyroid eQTL and *Inverse T2D-BP risk* cluster colocalized genes was performed using *metascape*[54]. For thyroid eQTL, we identified 244 colocalized genes and 75 SNVs in cCRE regions in follicular cells (used as major cell type within the thyroid). Among them, 15 signals were both associated with a colocalized gene in thyroid and a cCRE region in follicular cells. We used these remaining 15 colocalized genes with *metascape* default parameters. We performed the pathway analysis using 926 background genes expressed in the GTEx Thyroid RNA-seq v8 (TPM > 120). Similar analyses were conducted for the four other top colocalized tissues: adipose subcutaneous (adipocytes as major cell type), artery tibial (smooth muscle cells), nerve tibial (Schwann general cells), and skin (sun-exposed) lower leg (keratinocytes as major cell type). These analyses were also performed using *metascape*, with background genes expressed in GTEx RNA-seq v8 for each respective tissue: 1053 genes for adipose subcutaneous, 1053 for artery tibial, 1029 for nerve tibial, and 791 genes for skin (TPM > 120).

### scATAC-seq enrichment analysis

To gain insights in the associated regulatory processes, we took the 1304 clustered SNVs to look for enrichment of regions of open chromatin within clusters. Specifically, we aggregated independent SNVs (LD $r^2 < 0.2$) within 50 kb of each clustered SNV from the 1000 Genome Project Phase 3. We followed a similar protocol by the T2DGGI paper[9], and conducted a Firth bias-reduced logistic regression, using *logistf* R package and the following equation:

$$E(Y_i) = \alpha_0 + \alpha_{EXON} EXON_i + \alpha_{3UTR} 3UTR_i + \alpha_{5UTR} 5UTR_i + \sum_j \alpha_{ij} X_{ji}$$

With $Y_i$ taking the value 1 if the $i$-th SNV is within one of the clusters, $X_{ji}$ taking the value 1 if the $i$-th SNV mapped to an ATAC-seq peak for the $j$-th cell type, 0 otherwise. We defined $EXON_j$, $3UTR_j$, $5UTR_j$ indicators taking the value 1 if the $i$-th SNV is located to the respective annotation (from the Ensembl Project), 0 otherwise. The $\alpha$ are the coefficients of log fold enrichments, and $\alpha_0$ is the intercept.

We performed the logistic regression twice, one with $\alpha_{ij} = 0$ and one without constraint. We reported on the heatmap of Fig. 3b the $p$ value associated to Chi-squared tests between the two logistic regression models to observe the enrichment of the $j$-th cell type across clusters. We used multiple correction testing for Chi-squared tests across cell types and clusters, $p_{threshold} = 2.25 \times 10^{-4}$. More information on the 222 cell types gathered in CATLAS can be found in CATLAS website.

### Prevalence and relative risk of T2D-BP comorbidity using partitioned PGS

The prevalence of T2D-BP comorbidity was defined by the proportion of cases divided by the total number of individuals in the UKB, $Prev = \frac{N_{cases}}{N_{total}}$. The summary table in Fig. 4b provides the prevalence of individuals with T2D-BP comorbidity (of having both T2D and hypertension) based on being in the top decile (top 10%) of the partitioned unweighted PGS distribution. Additional results are available in Supplementary Table 5, showing both the top 10% and 33% of those partitioned PGS and comparing them with weighted T2D and SBP PGS. For individuals associated with multiple clusters, assignment was based on the cluster where they ranked the highest. The relative risk (RR) is calculated as the ratio of the proportion of cases within a cluster subgroup to the proportion of cases in the overall population $RR = \frac{Prev_{cases, cluster}}{Prev_{cases, overall}}$. The 95% confidence interval of the RR was estimated using the following equation:

$$CI_{95\%} = \exp\left[\ln(RR) \pm 1.96 \sqrt{\frac{1}{N_{cases, cluster}} - \frac{1}{N_{cases, overall}}}\right]$$

### Survival analysis using partitioned PGS

Survival analysis was conducted using the *survival* R package v3.6-4 with a Cox proportional hazards model. In this model, time zero was defined as the date of the first diagnosis of either hypertension or type 2 diabetes (T2D). The primary event of interest was the subsequent development of a T2D-BP comorbidity, marked by the diagnosis of the second disease. To ensure adequate sample sizes within groups, individuals were assigned to a cluster group if their maximum partitioned unweighted PGSs was in the top 33% of distribution. For individuals associated with multiple clusters, assignment was based on the cluster where they ranked the highest. Figure 4c displays the cumulative hazard with two-sided 95% confidence intervals. Sensitivity analysis includes comparing this survival analysis using T2D weighted PGS and SBP weighted PGS respectively, and is included in Supplementary Fig. 11 and Supplementary Data 10.

### Partitioned PGS and risk of complications

Following the same pipeline developed in *Reciprocal risk prediction using PGS*, partitioned PGSs were calculated using clustered SNVs in the risk-increasing direction of T2D, assuming an additive model (Supplementary Data 8). We do not associate a weight with partitioned PGSs, meaning each allele increasing the risk of T2D is counting as one. Sensitivity analysis included using weights derived from T2D and BP GWAS to predict the risk of the other disorders (Supplementary Data 9). Association with complications were performed using the UKB ICD-10 codes, updated with hospital data up to December 2023. Associations were corrected for covariates including age, genetically-inferred sex, genetic array, and the first six principal components. Briefly, we extracted all the UKB ICD-10 available codes and refined them to subcategories of interest, namely: Endocrine, nutritional, or metabolic; Mental and behavioural disorders; Nervous system, Eyes, ears, nose and throat; Circulatory system; Respiratory system; Digestive system; Skin; Musculoskeletal system; Genitourinary system. All reported associations in Fig. 5 were derived using unweighted partitioned PGS, *comorbidPGS* using binary logistic regression, and a P-value corrected for multiple testing, $p_{threshold} = 7.75 \times 10^{-5}$.

### Inclusion and ethics

This study is based exclusively on previously published and publicly available datasets; no new data involving human participants or animals were collected. Hence, no additional ethical approval was required for this research. We made use of multi-ancestry datasets to

the fullest extent possible. When robust data across multiple ancestry groups were available, all were included in the analysis. In cases where no suitable alternative was available, analyses were restricted to individuals of genetically inferred European ancestry. All references to ancestry are based on genetically inferred population structure and not on self-reported race or ethnicity. Similarly, the reported sex in this work was determined using genetic data.

## Reporting summary
Further information on research design is available in the Nature Portfolio Reporting Summary linked to this article.

## Data availability
The GWAS used in this study are all publicly available and listed in Supplementary Data 2. The UK Biobank Resource [https://ukbiobank.ac.uk/] was accessed using the Application Number 236. GTEx [https://www.gtexportal.org/home/downloads/adult-gtex/qtl] and TIGER eQTL data used in this study are publicly available via these links, respectively. Data from the ABOS cohort are not publicly available, since they are subject to national French data protection laws and restrictions imposed by the ethics committee to ensure data privacy of the study participants. Data can be accessed through an individual project agreement with the principal investigator of the University Hospital of Lille (Lille, France), F.P., using the email address: francois.pattou@univ-lille.fr. The ATAC-seq data from CATLAS are publicly available and can be accessed via the following link: https://catlas.org/humanenhancer/data/.

## Code availability
The software used for this analysis can be found on Zenodo [https://doi.org/10.5281/zenodo.17448298].

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

## Acknowledgements

This research has been conducted using the UK Biobank Resource under application number 236. This project was in part funded by the Agence Nationale de la Recherche under the Programme d'Investissement d'Avenir (PreciDIAB, ANR-18-IBHU-0001 and RHU PreciNASH ANR-16-RHUS-0006), by the European Union through the "Fonds Européen de Développement Regional" (FEDER), by the "Conseil Régional des Hauts-de-France" (Hauts-de-France Regional Council), by the "Métropole Européenne de Lille" (MEL, European Metropolis of Lille), and by the European Research Council (ERC OpiO – 101043671, to A.B.). I.Pr. and Z.B. were in part funded by the Diabetes UK (BDA number: 20/0006307), UKRI (EP/Z535072/1), European Foundation for the Study of Diabetes (EFSD), and Novo Nordisk A/S Programme for Diabetes Research in Europe—2025. P.M. acknowledges the support of the National Institute for Health and Care Research Barts Biomedical Research Centre (NIHR203330); a delivery partnership of Barts Health NHS Trust, Queen Mary University of London, St George's University Hospitals NHS Foundation Trust and St George's University of London. The research of Y.S. was funded by The British Academy (RaR\100084). The authors would like to thank all the investigators from different consortia that built and shared the GWAS meta-analysis, eQTLs, and scATAC-seq atlases used in this study, as well as the UK Biobank participants and dedicated staff.

## Author contributions

V.P. and I.Pr. designed the experiments and led the manuscript writing. L.Z. contributed to the study design for clustering and partitioned PGS. L.M. contributed to the analysis for colocalization, pathway analysis, interpretation, and revision. A.U., J.G.M., A.D., Z.B., I.Pu., and Y.S. defined the phenotypes of interest in the UK Biobank and provided handmade GWAS summary statistics. F.P. and B.S. provided the ABOS cohort. M.K. contributed to the overall statistics evaluation. A.K. contributed to the colocalization evaluation. V.P., A.B., P.M., P.F., and I.Pr. contributed to the evaluation of the results. P.F. and I.Pr. jointly supervised the study. All authors read and approved the final paper.

## Competing interests

V.P. is employed by Genomics Ltd., L.Z. is employed by Lifebit Biotech Inc. The authors declare that they have no other financial or non-financial competing interests. The views expressed in this study are the personal views of V.P. and L.Z., and do not represent the views of their current employers.
