## [Transparent Peer Review file · Nature Communications]

Partitioned polygenic scores show mechanistic heterogeneity in type 2 diabetes and hypertension comorbidity

Corresponding Author: Professor Inga Prokopenko

Version 0:

Reviewer comments:

Reviewer #1

(Remarks to the Author)

Pascal and colleagues present results of multi-layered analyses that use partitioned polygenic scores to understand heterogeneity in mechanisms driving type 2 diabetes (T2D) and hypertension comorbidity. They identify five clusters of SNPs associated with T2D and/or blood pressure traits that reflect different biological pathways to disease. Partitioned polygenic scores derived from these clusters improve prediction of T2D-hypertension comorbidity and offer a promising route to personalised prevention and management of these high-burden conditions. The manuscript tackles an extremely important and timely question related to conditions that drive global morbidity and mortality, which will be of broad interest to the readership of Nature Communications. The manuscript is generally clearly written (see some comments below), the methods applied are robust, and the interpretation of the findings are justified.

Comments.

1. I found it difficult to follow the GWAS datasets that were used for T2D and blood pressure traits at various steps in the analysis, and I think it would be useful in each results section to introduce the datasets used (including sample sizes, ancestry group). I'll outline my understanding below:

(a) If I understand correctly, the genetic correlation analyses use summary statistics for European ancestry individuals from Vujkovic et al, but was this from the Million Veteran Program alone, or combined with DIAGRAM Consortium results (as I believe both are available)? Similarly, for blood pressure, these analyses use summary statistics for European ancestry individuals from Evangelou et al, but did this include UK Biobank or not? These datasets are somewhat out of date as much larger studies have now been published with much larger samples sizes for both T2D and blood pressure, so it would be good to have clear motivation for their use.

(b) If I understand correctly, the summary statistics used to build the SNP x trait matrix for clustering are outlined in Supplementary Table 4, and these include T2D and blood pressure from Vujkovic et al. and Evangelou et al. This seems clear, although I note that Supplementary Table 4 indicates that the Vujkovic et al. results are from the DIAGRAM Consortium (which confused me). It was also be good to emphasize to the reader if T2D and the blood pressure traits were used for the clustering: the manuscript states that clustering was based on "49 related (endo)phenotypes" which might give the impression that they were not included. My confusion is then on where the list of 1,401 SNPs came from that were subsequently pruned and used as input to the clustering were obtained from. The manuscript indicates that they were curated from "multiple sources", but I think it is important to provide details of these sources. Importantly, did any of these sources included data from UK Biobank as this could bias any of the downstream testing of the partitioned polygenic score in UK Biobank.

(c) In testing the overall polygenic score in UK Biobank, effect sizes for T2D were taken from Vujkovic et al. (again, it was not clear whether the weights were obtained from summary statistics that included DIAGRAM) and for blood pressure were

taken from Warren et al. (rather than Evangelou et al.). Is there a reason that the Evangelou weights were not used? The Warren et al. summary statistics are based on UK Biobank, so this could bias testing of the partitioned polygenic score in UK Biobank.

2. In the clustering of SNPs, were the Z-scores used as input for each trait standardised by the sample size? If not, the clustering will be dominated by the (endo)phenotypes with the largest sample size.

3. The comparisons of clusters to previously reported clusters for T2D is very useful, and it is reassuring to see some overlap of features across clusters. Have the authors considered doing a similar comparison with previously reported clusters for hypertension (Vaura et al. 2022, *Circ Genom Precis Med* 15:e003583). They identify four different clusters using data from FinnGen and FINRISK, and it would be interesting to see how these overlap with clusters reported in this manuscript.

4. In testing the partitioned polygenic score with outcomes in UK Biobank, it was not clear what weights were being used in all analyses. For the analysis of complications, no weights were used (with sensitivity analyses using the T2D or blood pressure weights). However, what was done for the relative risk of the T2D and blood pressure comorbidity and the survival analysis.

5. For the survival analysis, I was not clear why individuals would need to be assigned to a cluster. Would it not be possible to simply fit a model using the five partitioned polygenic scores in a single model (or separate models) as predictors?

Reviewer #2

(Remarks to the Author)

Pascat and colleagues describe a study where they investigate the genetic basis of type 2 diabetes (T2D) and hypertension comorbidity using partitioned polygenic scores derived from clustered genetic variants. They analyze large-scale multiomic data to identify five distinct mechanistic clusters of genetic variants that contribute to T2D-hypertension risk, and demonstrate how these clusters can improve risk prediction for disease comorbidity. This is an interesting study but I do have some conceptual and methodological concerns as denoted below:

- Paragraph starting with line 129: I do not understand why metabolic syndrome is a separate cluster, as it is a collection of conditions that co-occur together (hence the name "syndrome"; including T2DM, HTN, hyperlipidemia etc.), typically linked to excess adiposity as the main driver. From a pathophysiology standpoint, I am not sure this makes logical sense, as opposed to including the main recognized driving factor, excess adiposity, as a cluster. As such, I am also having difficulty understanding how the finding of no association with BMI could be true (as explained in lines 147-149, with further inconsistent findings as explained in the paragraph starting with line 166 referring to the higher adiposity cluster), what is the authors' explanation for this finding?

- Paragraph starting with line 379: Why did the authors not identify independent loci (not just using the significance and LD threshold but also collapsing within a genomic distance such as 250 kilobases or 1 megabase). Also, am I understanding correctly that there were only 9 SNVs associated with both T2DM and BP traits? That seems to be very low, I would have expected more given the shared pathophysiology, mainly driven by excess adiposity. Looking at lines 104-106, I am more confused about this. The authors should clarify this point.

- Paragraph starting with line 415: The authors indicate using pre-computed LD scores from 1000 Genome Phase 3 European ancestry subset on multi-ancestry GWAS for T2DM and BP traits, which I suspect will impact the reliability of both SNP-based heritability, as well as genetic correlation between traits. How do the authors defend this analytical decision?

- Paragraph starting with Line 444: Given the multi-ancestral nature of T2D and BP GWAS, what reference panel did the authors utilize (specifically what ancestry group) in their pursuit of finding proxies via LD?

Reviewer #3

(Remarks to the Author)

I read with great interest the manuscript by Pascat et al., which aims to elucidate the shared genetic architecture of type 2 diabetes (T2D) and hypertension by overlaying existing GWAS datasets for the two conditions and identifying clusters of overlapping variants. The authors further attempt to demonstrate that these clusters may enhance genetic risk prediction for the comorbidity. While the study is thoughtfully conceived and methodologically sound, several aspects merit further consideration and clarification (not ranked by importance):

1. Given the well-established comorbidity between T2D and hypertension—likely driven by shared environmental and biological risk factors, most notably obesity—it is perhaps unsurprising that a non-zero genetic correlation exists between the two (e.g., even if they were genetically “distinct” but shared 1 risk factor that was itself partly genetic, they would genetically correlate). This has already been demonstrated in prior studies. The manuscript might benefit from acknowledging this expected correlation more explicitly and discussing how their clustering approach adds value beyond

confirming what is already previously studied.

2. It may be important for the authors to clarify the degree to which the respective GWAS datasets excluded individuals with the comorbid condition. For instance, if T2D GWAS cases include individuals with high blood pressure (which is probable), and vice versa, then the observed genetic overlap might in part reflect phenotypic correlations or sample contamination. A more detailed discussion—or at minimum, a caveat—regarding this potential bias may be warranted.
3. The authors' approach of selecting 1,401 genome-wide significant variants and clustering them before downstream annotation is reasonable. However, this top-hit strategy risks bias from the winner's curse and excludes a large proportion of potentially informative polygenic signal. Would it not be more useful to use the full genome-wide summary statistics and apply methods such as GNOVA, partitioned LDSC, or some more sophisticated multivariate GWAS approach (e.g. genomic SEM) to partition the shared architecture more comprehensively? While I am not suggesting the current approach be abandoned, it would be helpful for the authors to justify their methodological choices and contrast them with possible alternatives.
4. The use of a "comorbidPGS" is intriguing. However, the authors should acknowledge that this directly reflects the genetic correlation already demonstrated via LDSC. In other words, the predictive utility of one trait's PGS for the other trait reflects such genetic correlation. Although rescaling the LDSC genetic correlation into PGS's may offer some improved clinical utility, it would be helpful to frame these findings as extensions—rather than independent confirmations—of the same underlying genetic relationship.
5. The observation that cluster-informed PGSs predict comorbid outcomes is novel, but perhaps expected, given that these clusters were defined based on shared signal. To strengthen their conclusion that such partitioned clusters offer clinical value, the authors should compare these against the individual PGSs for T2D and BP, and ideally also a combined PGS (e.g., using genomic SEM-derived weights). Demonstrating that the cluster-based approach offers superior predictive accuracy over such approaches would substantiate the claim that their cluster-partitioned PGSs are more clinically useful.
6. The statement that their approach enables "identifying younger individuals at high risk more effectively" is difficult to support using UKB, which is limited to middle-aged and older adults. A more cautious phrasing is advised, unless the authors are able to validate their findings in younger cohorts (e.g., birth cohorts with longitudinal follow-up). The current data do not allow conclusions about predictive performance in truly young populations. Furthermore, the existing GWAS data is primarily from late-adult/mid-adult populations, but the ongoing "lifecourse GWAS" may help answer such questions in the future.
7. The manuscript references the mitigation of "... pleiotropy effects" (row 329) but the meaning of this is unclear. If the authors refer to horizontal pleiotropy between T2D and BP, it is not evident how their method—especially one that selects top variants—would control for this. It is also not clear how such pleiotropy would affect their findings (if at all??). Some clarification or rewording is needed to prevent conceptual ambiguity.
8. The identification of a cluster associated with "Inverse T2D-BP risk" is certainly intriguing. However, such a pattern could emerge naturally given incomplete overlap in genetic architectures. Mathematically, one would expect such clusters to exist in the symmetric difference between genetic variance of each condition. While still important to highlight, the biological interpretation should be handled cautiously.
9. The authors may wish to acknowledge more explicitly that their findings are limited to common variants. Extending this clustering framework to include whole-exome or whole-genome—should such data become available—could offer additional insights and deserves mention as a future direction.
10. The conclusions in the final sections overstate the demonstrated findings. In particular, the claim of "improved accuracy" and the suggestion that this work provides "a foundation for precision medicine" are premature without direct comparison to conventional PGS strategies (the individual ones or a combined PGS, see point 5) or demonstration of direct clinical utility. A more tempered conclusion would better reflect the limitations and incremental nature of this work.
11. I am somewhat concerned about the use of GWASs data from different ancestries. While I share the authors enthusiasm to increase diversity in data, most methods employed assume a fixed or common LD structure – and combining data from across ancestries would therefore introduce bias via population stratification. Please consider the implications of this and amend accordingly. I note that the authors correctly restricted their LDSC to European ancestry.

Overall, this is an ambitious and promising study. With some refinement and additional contextualization, it could make a valuable contribution to the literature on polygenic architecture and risk prediction for comorbid cardiometabolic diseases.

Version 1:

Reviewer comments:

Reviewer #1

(Remarks to the Author)

The authors have thoughtfully considered my comments and have conducted additional analyses to address the concerns I have raised. The manuscript is now much clearer with regard to the datasets used for each of the different analyses, and these are now better motivated in the main text. I have two remaining minor suggestions:

Previous comment 3 suggesting a comparison with previously reported BP clusters. The authors have dealt with this comment well. However, I wonder if it would be better to present to comparisons between clusters in Supplementary Figure 6 (and also in Supplementary Figure 5) using a Sankey plot. I believe this more clearly shows how the clusters are related to each other.

Previous comment 5 about the survival analysis. I accept the authors argument here, but I wonder if this should be emphasized in the main text (maybe discussion) as the points they have made are important.

Reviewer #2

(Remarks to the Author)

Authors present a very detailed rebuttal and have answered my initial questions/concerns regarding the manuscript.

Reviewer #3

(Remarks to the Author)

The authors have satisfactorily addresses my comments. Congratulations on a nice contribution.

AUTHOR REBUTTAL LETTER

Dear Editor and Reviewers,

We are very grateful to the reviewers, editor and whole editorial team for their time and thoughtful suggestions.

We carefully addressed the reviewers' comments which we believe have substantially improved our manuscript about the mechanistic heterogeneity between high Blood Pressure (BP) and Type 2 Diabetes (T2D) using Polygenic Score (PGS).

In the below document we provide responses to the review comments point-by-point, using **bold blue text** to highlight the reviewer's question and plain black text for our response. We also provide citations of the introduced changes in *italic text, with added words in red*.

We **highlighted in yellow** the changes introduced to the main text and followed other instructions for the submission.

Reviewer Comments

Reviewer #1 (Remarks to the Author):

Pascat and colleagues present results of multi-layered analyses that use partitioned polygenic scores to understand heterogeneity in mechanisms driving type 2 diabetes (T2D) and hypertension comorbidity. They identify five clusters of SNPs associated with T2D and/or blood pressure traits that reflect different biological pathways to disease. Partitioned polygenic scores derived from these clusters improve prediction of T2D-hypertension comorbidity and offer a promising route to personalised prevention and management of these high-burden conditions. The manuscript tackles an extremely important and timely question related to conditions that drive global morbidity and mortality, which will be of broad interest to the readership of Nature Communications. The manuscript is generally clearly written (see some comments below), the methods applied are robust, and the interpretation of the findings are justified.

Comments.

1. I found it difficult to follow the GWAS datasets that were used for T2D and blood

pressure traits at various steps in the analysis, and I think it would be useful in each results section to introduce the datasets used (including sample sizes, ancestry group). I'll outline my understanding below:

(a) If I understand correctly, the genetic correlation analyses use summary statistics for European ancestry individuals from Vujkovic et al, but was this from the Million Veteran Program alone, or combined with DIAGRAM Consortium results (as I believe both are available)? Similarly, for blood pressure, these analyses use summary statistics for European ancestry individuals from Evangelou et al, but did this include UK Biobank or not? These datasets are somewhat out of date as much larger studies have now been published with much larger samples sizes for both T2D and blood pressure, so it would be good to have clear motivation for their use.

We would like to thank the reviewer for this observation. Given the multiple datasets cited in the manuscript, we updated **Supplementary Table 4** to provide a complete list of all GWAS summary statistics used, along with details on whether UK Biobank (UKB) participants were included.

In this study, we used a set of two GWAS for both T2D and BP traits, one including UKB individuals – used as the most powerful GWAS for the main analyses such as hierarchical clustering, colocalisation. For analyses where sample overlap could bias results such as LD score regression and the construction of comorbid PGSs, we intentionally used European-ancestry GWAS summary statistics that excluded UKB participants. To ensure consistency across our work, we decided to include somewhat older GWAS for T2D/BP traits since they had a version without UKB individuals. Specifically, for T2D, we used Mahajan et al. dataset¹ with or without UKB individuals, available via the DIAMANTE consortium website <https://diagram-consortium.org/downloads.html>. We used Warren et al.² data which was the biggest GWAS available to us for BP traits. To run analysis on the UKB, we obtained the summary statistics for a subset, which also excludes UKB individuals, from the authors of Evangelou and colleagues³ from the International Consortium of Blood Pressure (ICBP) GWAS.

We acknowledge that larger GWASs have since been released (e.g., Suzuki et al. for T2D⁴ and Keaton et al. for BP⁵), but at the time of curation (June 2023), these newer datasets were either not yet publicly available or lacked sufficient documentation to allow us to replicate the UKB-exclusion strategy. We now clarify this point in the **Methods** section in line 407:

“Particularly for T2D, we used Mahajan et al.¹ GWAS (2018b) as it allows us to alternate between a GWAS with UKB individuals or not, accessible on the DIAGRAM/DIAMANTE/T2DGGI webpage <https://diagram-consortium.org/downloads.html>. We alternated between using the Warren et al. GWAS² based on the UKB and a subset version of the Evangelou et al. GWAS³ without UKB (only ICBP data). Both T2D/BP GWASs without UKB individuals were used only for two specific analyses: LD score regression and PGS.”

Additionally, we added references to the datasets used on lines 443 and 452 respectively:

Line 456:

“Genetic correlation. We conducted genetic correlation analysis by using the whole T2D¹ and BP³ GWAS datasets of European ancestry only without UKB individuals.”

Line 464:

“Reciprocal risk prediction using PGS. We constructed PGS before and after clustering with plink v1.9⁶, using specific SNV sets, weights from T2D¹, SBP, DBP, and PP³ summary statistics (GWAS without UKB individuals) and individual-level data from the UKB.”

(b) If I understand correctly, the summary statistics used to build the SNP x trait matrix for clustering are outlined in Supplementary Table 4, and these include T2D and blood pressure from Vujkovic et al. and Evangelou et al. This seems clear, although I note that Supplementary Table 4 indicates that the Vujkovic et al. results are from the DIAGRAM Consortium (which confused me). It was also be good to emphasize to the reader if T2D and the blood pressure traits were used for the clustering: the manuscript states that clustering was based on “49 related (endo)phenotypes” which might give the impression that they were not included. My confusion is then on where the list of 1,401 SNPs came from that were subsequently pruned and used as input to the clustering were obtained from. The manuscript indicates that they were curated from “multiple sources”, but I think it is important to provide details of these sources. Importantly, did any of these sources included data from UK Biobank as this could bias any of the downstream testing of the partitioned polygenic score in UK Biobank.

We are thankful to the reviewer for pointing out this comment. We apologise for the confusion as we mentioned Vujkovic et al.⁷ in the reference, but only Mahajan et al.¹ with and without UKB individuals was used in this analysis. We diligently corrected this in the manuscript with references, and added a paragraph in the methods section to clarify this point line 407:

“Particularly for T2D, we used Mahajan et al.¹ GWAS (2018b) as it allows us to alternate between a GWAS with UKB individuals or not, accessible on the DIAGRAM/DIAMANTE/T2DGGI webpage <https://diagram-consortium.org/downloads.html>.”

Secondly, we added a paragraph in both the Results and the Methods section associated with the hierarchical clustering, as both T2D- and BP-related traits were used in the clustering. With this study design, we wanted to clearly capture shared genetic variants defining susceptibility to different mechanisms in T2D-BP comorbidity, as such, these traits were included in the clustering.

Results

Line 118:

“Clusters of pathogenetic processes. To dissect the complexity of shared biological pathways between T2D and BP, we curated and refined the SNV list to 1,304 independent variants (LD $r^2 < 0.2$), including 500 T2D-associated and 813 BP-associated SNVs, to cluster them based on their effects on 45 related (endo)phenotypes including T2D/BP (Supplementary Table 4).”

Methods

Line 486:

“Clusters of pathogenetic processes. We aggregated the 1,304 independent T2D-BP SNVs and investigated 45 endophenotypes GWAS summary statistics including

T2D/BP (Supplementary Table 4) to cluster them into distinct groups based on pathogenetic processes.”

We thank the reviewer for this remark on the origin of our SNVs associated with T2D and BP traits. We completed the **Methods** section by clarifying the sources from which these SNVs were drawn. Specifically for T2D, we used GWAS prior to Mahajan 2018b⁸⁻¹². We used a similar strategy for BP traits, compiling SNVs reported for any of the three phenotypes (SBP/DBP/PP)¹³⁻²⁵. To minimise bias, we prioritised studies that did not include the UK Biobank or only incorporated it as part of a meta-analysis, in which case we favoured signals identified in the non-UK Biobank component.

Line 413:

“To ensure robustness in uncovering shared aetiologies, we gathered from multiple sources a list of T2D⁸⁻¹² and BP SNVs¹³⁻²⁵ reaching genome-wide significance (i.e., P-value < 5×10⁻⁸) and characterised by prevalence (minor allele frequency (MAF) > 0.01). SNV could have been reported for either one of the three BP metrics (SBP/DBP/PP) or it was reported for T2D”

(c) In testing the overall polygenic score in UK Biobank, effect sizes for T2D were taken from Vujkovic et al. (again, it was not clear whether the weights were obtained from summary statistics that included DIAGRAM) and for blood pressure were taken from Warren et al. (rather than Evangelou et al.). Is there a reason that the Evangelou weights were not used? The Warren et al. summary statistics are based on UK Biobank, so this could bias testing of the partitioned polygenic score in UK Biobank.

We thank the reviewer for this observation. As mentioned in the response to reviewer’s previous comment, we mainly used Warren et al.² dataset, which was the biggest GWAS available to us for BP traits. However, to construct PGS using UKB as target dataset or estimate the genetic correlation between T2D and BP, we used Evangelou et al.³ weights without UKB individuals for BP PGS, and Mahajan et al.¹ without UKB individuals for T2D PGS. We corrected our references and added a section in the **Methods** to describe T2D/BP datasets:

Line 410:

“We alternated between using the Warren et al. GWAS² based on the UKB and a subset version of the Evangelou et al. GWAS³ without UKB (only ICBP data). Both T2D/BP GWASs without UKB individuals were used only for two specific analyses: LD score regression and PGS.”

2. In the clustering of SNPs, were the Z-scores used as input for each trait standardised by the sample size? If not, the clustering will be dominated by the (endo)phenotypes with the largest sample size.

We thank the reviewer for raising this important methodological point. To assess whether our clustering reflects true shared genetic architecture, we conducted an extensive comparison of two preprocessing strategies:

- Unadjusted Z-scores (as in the original submission), which scales with \sqrt{N} and therefore inherently weights stronger more precisely estimated associations more heavily.
- Sample-size-standardized Z-scores, obtained by dividing each Z by \sqrt{N} , to place all traits on an equal footing regardless of cohort size.

We acknowledge the trade-off between these two approaches. On one hand, unadjusted Z-scores prioritise high-confidence associations which simultaneously encodes effect size and variance with each endophenotype in a comorbidity framework involving two traits. On the other hand, sample-size normalisation improves comparability across GWASs with heterogeneous sample sizes. In our case, both T2D and BP traits were jointly used in the clustering; we chose to retain the unadjusted Z-scores because they emphasize variants most confidently implicated in at least one of the two conditions. This is particularly important as we are addressing genetic comorbidity, which may involve heterogeneity, not present when clustering within a single disease (e.g. as in the T2D clustering by Suzuki et al.⁴).

We acknowledge other studies such as Suzuki et al.⁴ did use Z-score adjusted for sample sizes; however, those studies were not tackling the comorbidity between two diseases, which may lead to higher heterogeneity in variant effects on related endophenotypes than clustering T2D signals alone.

To empirically evaluate the robustness of our approach, we now include **Supplementary Figure 2**, which compares cluster assignments from both methods. The resulting clusters demonstrate substantial concordance, and the overall biological conclusions remain unchanged. To quantify this similarity, we provide a **contingency heatmap (Supplementary Figure 2c)** of variant assignments across both clustering approaches. We additionally performed a **chi-squared test of independence**, which showed a strong non-random overlap between cluster labels ($p < 2 \times 10^{-16}$), hence supporting the stability of the cluster structure. We added this analysis in the manuscript in the Results and the Methods sections respectively.

Results

Line 132:

“To ensure robustness of our clustering, we ran extensive sensitivity analyses (Methods) with different sets of metabolic traits and other clustering methods, such as using Z-score adjusted for GWAS sample size, MRClust²⁶ and Bayesian nonnegative matrix factorization (bNMF) (Methods, Supplementary Figure 1, 2, 3, and 4).”

Methods

Line 523:

“Additional sensitivity analyses include the use of three alternative clustering methods: adjusting Z-score for GWAS sample size prior to clustering (using the following formula $Z_{ij} = \frac{\beta_{ij}}{\sigma_{ij} \times N_j}$, Supplementary Figure 2), MRClust²⁷ (Supplementary Figure 3) and a ‘soft clustering’ method, bNMF²⁸ (Supplementary Figure 4). In Supplementary Figure 2a and b, we showed a comparative hierarchical clustering using our approach without GWAS sample size adjustment, and another approach by adjusting for sample size (dividing Z by \sqrt{N} , N being the sample size for a given GWAS), as it has been done in other studies⁴. We decided to use unadjusted Z-scores as main analysis as we wanted to prioritise robust, high-confidence

associations. We showed in **Supplementary Figure 2c** that the clusters assignment has followed the same patterns with or without adjustment (Chi-squared test of independence $p < 2 \times 10^{-16}$).

Figure 1. The added Supplementary Figure 2 shows an extensive comparison of two preprocessing strategies (adjusting Z scores for sample size or not). a. Hierarchical clustering with unadjusted Z-scores (original figure 2); b. Hierarchical clustering with adjusted Z-scores; c. Contingency heatmap of the assignment of the 1,304 variants in the non-adjusted vs adjusted clustering (Chi-squared test of independence $p < 2 \times 10^{-16}$).

3. The comparisons of clusters to previously reported clusters for T2D is very useful, and it is reassuring to see some overlap of features across clusters. Have the authors considered doing a similar comparison with previously reported clusters for

hypertension (Vaura et al. 2022, Circ Genom Precis Med 15:e003583). They identify four different clusters using data from FinnGen and FINRISK, and it would be interesting to see how these overlap with clusters reported in this manuscript.

We sincerely thank Reviewer for their suggestion. Following this recommendation, we compared our clustered SNVs with those reported by Vaura and colleagues²⁹, who identified four hypertension-related clusters in the FinnGen and FINRISK cohorts.

Specifically, we used SNVs from Vaura et al.²⁹ that were in linkage disequilibrium with our clustered variants ($r^2 > 0.6$ in European populations) and integrated these overlaps in **Supplementary Table 5**. In addition, we provide a new **Supplementary Figure 6**, which visualizes the distribution of the overlapping/correlated variants across our clusters.

Despite the relatively small number of overlapping variants (N=19), we can identify some emerging patterns. Variants associated with obesity-related traits in Vaura et al.²⁹ predominantly mapped to our *Higher adiposity* cluster. Variants related to short stature aligned with our *Inverse T2D-BP risk* cluster, which features short stature. Variants from their hypolipidaemia cluster were enriched within our mechanistically similar *Vascular dysfunction* cluster. As anticipated, we observed greater variability in the mapping of shared variants within our *Metabolic syndrome* cluster, consistent with its heterogeneous nature.

In addition to the table and figure, we added information into **Results** and **Method** sections to include this additional analysis.

Results:

Line 137:

*“The pathophysiological processes identified across the five clusters were consistent with existing evidence and highlight novel mechanistic insights (**Methods, Supplementary Figure 4 to 7**)^{3,4,30}.”*

Line 169:

*“Comparison with previously reported BP-related clusters showed partial overlap with the Hypolipidaemia and Short stature clusters (**Supplementary Figure 6**)²⁹.”*

Line 176:

*“The **Vascular Dysfunction cluster** includes 287 SNVs mostly originating as BP signals. They are associated with cardiovascular traits (higher risk of AF, stroke, CAD, heart failure), lower birth weight and show strong effects on both T2D – high BP. This cluster showed a number of hypolipidemia SNVs from previous BP clustering (**Supplementary Figure 6**).”*

Methods:

Line 508:

*“To validate our clusters, we compared them with those identified in the latest T2DGGI study⁴ and both latest ‘soft’ clustering for T2D³⁰ and BP²⁹. First, we looked at SNV assignment between our T2D-BP clusters, T2DGGI clusters (**Supplementary Figure 5a**), latest T2D ‘soft’ clusters (**Supplementary Figure 5b**), and latest BP ‘soft’ clusters (**Supplementary Figure 6, Supplementary Table 5**). To do so, we systematically looked for LD proxies (LD $r^2 > 0.6$) in **the three clusters**. We assign a SNV into ‘soft’ clusters based on weight > 0.75 (**Supplementary Table 5, Supplementary Figure 5-6**).”*

Figure 2 : note in this plot, due to a very small number of variants in Vaura et al., we are not showing the unattributed variants among our clusters

4. In testing the partitioned polygenic score with outcomes in UK Biobank, it was not clear what weights were being used in all analyses. For the analysis of complications, no weights were used (with sensitivity analyses using the T2D or blood pressure weights). However, what was done for the relative risk of the T2D and blood pressure comorbidity and the survival analysis.

We thank the reviewer for raising this item. We used unweighted partitioned polygenic score based on the clusters for all main analyses in the UK Biobank post-clustering. We set unweighted partitioned PGS aligned to the T2D risk allele, *i.e.*, each allele increasing the risk of T2D is counted as one in the PGS calculation. This is in line with the fact that weighting for each variant would be different between T2D and the three BP traits. We used those unweighted PGS to evaluate comorbidity between T2D and hypertension (defined as an outcome by either SBP \geq 150 mmHg, or DBP \geq 90 mmHg, or taking blood pressure lowering medication, **Supplementary Figure 9**). We used the same unweighted PGS in survival analysis.

Clarifying this point, we mention “unweighted” alongside clarification in hypertension definition in the **Methods** section. We also amended the **Results** section.

Methods:

Line 431:

*“We defined the outcome “hypertension” used in the last section of the **Results** (unweighted PGS evaluation) as either having SBP \geq 150 mmHg, or DBP \geq 90 mmHg, or taking blood pressure lowering medication. The complete criteria used to identify individuals with T2D and hypertension are illustrated in **Supplementary Figure 10**. We identified 217,599 hypertensive individuals in the UKB (**Supplementary Table 1**).”*

Line 602:

*“The summary table in **Figure 4b** provides the prevalence of individuals with T2D-BP comorbidity (of having both T2D and hypertension) based on being in the top decile (top 10%) of the partitioned **unweighted** PGS distribution.”*

Line 618:

*“To ensure adequate sample sizes within groups, individuals were assigned to a cluster group if their maximum partitioned **unweighted** PGSs was in the top 33% of distribution.”*

Results:

Line 263:

*“**T2D-BP comorbidity using partitioned PGSs**. To evaluate the ability of the T2D-BP SNV clusters to predict comorbidities and complications, we used the individual-level data from UKB and built **unweighted partitioned PGSs for each of cluster group**. Using the R software environment tool, **comorbidPGS**, we aligned partitioned PGS to the T2D risk allele, i.e., each allele increasing the risk of T2D is counted as one in the PGS calculation (**Methods**)³¹.*

*To illustrate the influence of genetic predisposition, we computed the relative risk of comorbidity among the UKB individuals (**Methods**), based on the top 10% percentiles of each **unweighted** partitioned PGS.”*

5. For the survival analysis, I was not clear why individuals would need to be assigned to a cluster. Would it not be possible to simply fit a model using the five partitioned polygenic scores in a single model (or separate models) as predictors?

We thank the reviewer for this insightful suggestion. We agree that, statistically, one could fit a survival model using the five partitioned PGS as continuous predictors, either jointly or separately, to assess their independent contributions to risk.

However, in designing our analysis, our objective was to consider potential applications at the clinical level. We therefore assigned individuals to the cluster corresponding to their highest partitioned PGS cluster to simulate a more actionable framework, where an individual could be stratified into a most important mechanistic pathway for targeted risk prediction and management.

We fully acknowledge that the current partitioned PGSs are not yet suitable for direct clinical implementation. They are built from a limited set of predictors and do not match the performance of genome-wide approaches using robust methods such as LDpred²³² or PRS-

CS³³. Nevertheless, the goal of this assignment was to explore how future, more powerful versions of cluster-specific PGSs could be used in practice—for example, by defining thresholds to classify individuals into distinct risk profiles/signatures and guide screening strategies accordingly.

This vision aligns with precision medicine approaches already demonstrated through tools such as QDiabetes³⁴ and QRisk³⁵, which stratify individuals into risk categories based on clinical and demographic data. We believe that genetic partitioning could eventually complement such tools to enable earlier and more precise interventions, as it is done by the pilot 'Healthcare Evaluation of Absolute Risk Testing' (HEART) in the United Kingdom³⁶.

Reviewer #2 (Remarks to the Author):

Pascat and colleagues describe a study where they investigate the genetic basis of type 2 diabetes (T2D) and hypertension comorbidity using partitioned polygenic scores derived from clustered genetic variants. They analyze large-scale multiomic data to identify five distinct mechanistic clusters of genetic variants that contribute to T2D-hypertension risk, and demonstrate how these clusters can improve risk prediction for disease comorbidity. This is an interesting study but I do have some conceptual and methodological concerns as denoted below:

- Paragraph starting with line 129: I do not understand why metabolic syndrome is a separate cluster, as it is a collection of conditions that co-occur together (hence the name "syndrome"; including T2DM, HTN, hyperlipidemia etc.), typically linked to excess adiposity as the main driver. From a pathophysiology standpoint, I am not sure this makes logical sense, as opposed to including the main recognized driving factor, excess adiposity, as a cluster. As such, I am also having difficulty understanding how the finding of no association with BMI could be true (as explained in lines 147-149, with further inconsistent findings as explained in the paragraph starting with line 166 referring to the higher adiposity cluster), what is the authors' explanation for this finding?

We thank reviewer for bringing up the discussion about Metabolic Syndrome (MetS). In our manuscript, we labelled this cluster based on the observed joint effects of its variants on multiple cardiometabolic traits – including hypertension, T2D, central obesity and dyslipidaemia – which align with the composite criteria outlined recently by the International Diabetes Federation (IDF)³⁷.

The IDF definition of MetS is primarily focussed on excess of central obesity, measured by the waist circumference or waist-to-hip ratio (WHR), as a core component, accompanied by at least two of the following: dyslipidaemia, impaired glucose regulation, or high blood pressure. Importantly, WHR has been shown to more accurately capture visceral fat, which is metabolically active and contributes to insulin resistance, lipid abnormality, and inflammation – mechanisms central to the development of MetS. In contrast, BMI does not distinguish between muscle and fat, nor does it reflect fat distribution.

While higher overall body adiposity is a recognised risk factor for the vast majority of diseases^{38,39}, there is raising awareness that central obesity (proxied by WHR) at normal BMI plays even more important role in cardiometabolic risk. For example, individuals with normal BMI but central obesity have significantly higher odds of hypertension, T2D, and other cardiometabolic risks (CAD, stroke etc); WHR and waist circumference are independent predictors of cardiometabolic risk even after adjusting for BMI^{40–44}.

Moreover, recent MR studies reported causality of WHR (adjusted for BMI) on risk of insulin resistance (Lagou et al., Nat Comms, 2021, by co-authors of this study)⁴⁵ and provided causal support to previous epidemiological studies on WHR effect on T2D risk⁴⁶. This suggests WHR captures risk pathways that BMI misses.

Thus, the apparent absence of an effect on BMI in our MetS cluster likely reflects its specificity for central adiposity pathways, as captured by WHR and not BMI. We added in the **Discussion** the following paragraph.

Line 352:

“The Metabolic Syndrome cluster was associated with novel features such as shorter stature, higher WHR, and no detectable effect on BMI. These features are consistent with the International Diabetes Federation (IDF) definition of MetS³⁷, which emphasises central obesity – better captured by WHR than BMI – as the primary driver. The absence of an effect on BMI in this cluster likely reflects the specific contribution of visceral adiposity to cardiometabolic risk, rather than overall body mass, highlighting the relevance of WHR-linked pathways in MetS pathophysiology.”

We agree with the reviewer that MetS is a collection of conditions, and has historically been difficult to define genetically due to its heterogeneous definitions. This limited the success of GWAS approaches using MetS as a composite endpoint. However, our unsupervised clustering approach recovered a group of variants with shared effects consistent with central adiposity driven MetS. This data-driven framework supports the IDF MetS definition and could potentially bring future consensus in its genetic underpinnings in terms of symptoms/phenotypes.

Finally, we also found an overall *Higher adiposity* cluster, distinct from the MetS. This cluster shows strong association with BMI alongside a different pattern of associations. We clarified this distinction in the revised manuscript **Results** section.

Line 161:

“The Metabolic Syndrome cluster included 215 variants, and displayed the most distinct pathogenetic signature. It highlights attributes consistent with the metabolic syndrome, including lower levels of sex hormones (sex-hormone binding globulin, insulin, testosterone)^{47,48}, increased central adiposity (waist-to-hip ratio [WHR] adjusted for body-mass index [BMI])⁴⁹ without higher overall adiposity, measured by BMI, systemic higher IR evaluated by the homeostasis model assessment of insulin resistance, HOMA-IR, using both fasting plasma glucose and insulin (alongside higher HOMA-B, proinsulin level, and insulin fold change), lower high-density lipoprotein (HDL) cholesterol, higher triglycerides (TG), and altered cardiovascular functions (higher heart rate, increased cardiovascular event risk, higher renin-angiotensin-aldosterone system activity)⁵⁰.”

Line 191:

“The Higher adiposity cluster contained 137 SNVs – predominantly T2D signals – which showcased effects on higher BMI, reduced sex hormones (testosterone and SHBG), higher TG along with lower HDL- and LDL-cholesterol, higher risk of cardiovascular events (CAD, heart rate, stroke), and insulin resistance (higher HOMA-IR/HOMA-B). This cluster, distinct from the Metabolic Syndrome one, showed a high number of obesity-related SNVs within previous T2D clustering (Supplementary Figure 5).”

- Paragraph starting with line 379: Why did the authors not identify independent loci (not just using the significance and LD threshold but also collapsing within a genomic distance such as 250 kilobases or 1 megabase). Also, am I understanding correctly that there were only 9 SNVs associated with both T2DM and BP traits? That seems to be very low, I would have expected more given the shared pathophysiology, mainly driven by excess adiposity. Looking at lines 104-106, I am more confused about this. The authors should clarify this point.

We thank the reviewer for this insightful comment, which prompted us to further clarify this point in the revised manuscript.

First, we confirm that the 1,304 SNVs included in the clustering analysis were pruned for independence, applying the pairwise linkage disequilibrium (LD) threshold of $R^2 < 0.2$, based on European reference populations. We did not apply an additional genomic distance filter (e.g., collapsing within 250 kilobases or 1 megabase) during this pruning step. Our rationale for this choice was to preserve the potential for capturing variants with multiphenotype effects and local correlation structures that may contribute to the shared genetic architecture of T2D and BP traits. By relying solely on LD pruning, we reduced redundancy from strongly correlated signals while retaining biologically meaningful overlap which provides insights in the clustering for understanding comorbidity patterns. We acknowledge this could be criticised, and explicitly clarified this in the **Methods** section.

Line 416:

“We identified a total of 1,401 SNVs. This list was further curated for clustering to keep only independent SNVs (LD $r^2 < 0.2$). We did not apply additional pruning based on genetic distance. This decision was made to preserve the potential for capturing pleiotropic variants and regional architectures that could contribute to comorbidity between T2D and BP traits. The resulting list encompasses 1,304 SNVs, including 500 T2D and 283/272/270 SBP/DBP/PP signals, with 9 unique SNVs demonstrating associations with T2D-BP comorbidity.”

Second, we clarified the genetic overlap analysis in both **Results** and **Methods** sections. We indeed identified nine SNVs as lead signals in both T2D and high BP GWAS summary statistics (i.e., exact same variant as top signals in both T2D and high BP traits), but we also listed a further 97 SNV pairs overlapping among the 1,401 pre-curated non-independent list of SNVs reported for either T2D or high BP.

Results

Line 103:

*“We gathered a collection of 1,401 SNVs associated with T2D, high SBP, DBP, and/or PP (**Methods**). We revealed 24/19/26 genetic loci overlapping between T2D and SBP/DBP/PP respectively, determined by LD and/or genomic proximity (**Supplementary Table 3**). Of the 1,401 SNVs, 9 were directly reported as lead signals for both T2D and BP traits. Additionally, we identified 97 SNV pairs that overlapped (within 500 kb or LD $r^2 > 0.2$) and were associated with T2D and BP traits.”*

Methods

Line 481:

*“**Genetic overlap.** We estimated the genetic overlap among the pre-curated non-independent 1,401 SNVs by identifying variants within 500 kb of each other, or in LD $r^2 > 0.2$ for individuals of European ancestry. The reported loci are the proximal ones, identified by biomaRt⁶¹. We identified and coloured the signals per clusters in **Supplementary Table 3.**”*

While we agree these numbers may still seem low given the known shared pathophysiology, notably related to adiposity. However, we selected only SNVs that reached genome-wide significance ($P < 5 \times 10^{-8}$) for either T2D or BP regulation traits. As such, we are capturing only the extreme tail of the association distributions – variants with the strongest, most confidently replicated signals for each trait individually. A recent study proved that GWAS

study design is ranking signals by disease specificity instead of importance, which is also confirming that we are using material that should not highly overlap even if the disease architectures themselves do⁵². While the broader genetic architectures of T2D and BP are indeed correlated (as confirmed by our LD score regression analyses showing a significant genetic correlation up to 0.2 between SBP and T2D), the intersection of genome-wide significant variants between the two traits is, by statistical expectation, small. This is a direct consequence of the polygenic nature of both traits, where risk is spread across thousands of variants with small effects, and the stringent P-value threshold we used to minimise false positives.

- Paragraph starting with line 415: The authors indicate using pre-computed LD scores from 1000 Genome Phase 3 European ancestry subset on multi-ancestry GWAS for T2DM and BP traits, which I suspect will impact the reliability of both SNP-based heritability, as well as genetic correlation between traits. How do the authors defend this analytical decision?

We thank the reviewer for this important comment, which gave us the opportunity to clarify this point.

As indicated in the **Methods**, for the purpose of SNP-based heritability and genetic correlation analyses using *ldsc*, we restricted our analyses to subsets of GWAS summary statistics comprising only individuals of European ancestry. We chose to restrict our datasets to ensure compatibility with the precomputed LD scores from the 1000 Genomes Project Phase 3 European panel, thereby maintaining methodological consistency and avoiding potential biases introduced by ancestry mismatches. We clarified this GWAS selection point in the **Methods** section.

Line 402:

“Material description. We collected 45 publicly available GWAS summary statistics (Supplementary Table 4), spanning from 2017 to 09 June 2023. GWAS were selected if they included a large number of participants ($N \geq 10,000$) and a preference for datasets with diverse ancestral backgrounds, involving the majority of individuals of European ancestry, to enhance genetic diversity in our study. European-only studies were included when multi-ancestry data were unavailable. Particularly for T2D, we used Mahajan et al.¹ GWAS (2018b) as it allows us to alternate between a GWAS with UKB individuals or not, accessible on the DIAGRAM/DIAMANTE/T2DGGI webpage <https://diagram-consortium.org/downloads.html>. We alternated between using the Warren et al. GWAS² based on the UKB and a subset version of the Evangelou et al. GWAS³ without UKB (only ICBP data). Both T2D/BP GWASs without UKB individuals were used only for two specific analyses: LD score regression and PGS.”

While for other parts of the study, such as clustering, we leveraged larger, multi-ancestry GWAS datasets including a majority of European individuals to maximise power, we carefully adjusted our analyses depending on the requirements of each method. We however acknowledge that we used LD reference panel from 1000G of European ancestry for proxy finding, which is the content of the reviewer’s next comment.

- Paragraph starting with Line 444: Given the multi-ancestral nature of T2D and BP GWAS, what reference panel did the authors utilize (specifically what ancestry group) in their pursuit of finding proxies via LD?

We thank the reviewer for this comment, which allowed us to clarify our methodological choices in the revised manuscript.

Throughout the hierarchical clustering of this study, we faced a trade-off between maximizing power and generalisability – by leveraging the largest available GWAS datasets for the examined endophenotypes – and minimising potential bias due to population heterogeneity. To strike this balance, we prioritised GWAS datasets that were either exclusively European ancestry or multi-ancestry studies in which European ancestry individuals constituted the majority of the sample.

For LD proxy search in the hierarchical clustering, when a given SNV was not available in a particular GWAS dataset, we systematically used a European ancestry reference panel – specifically, the European subset of the 1000 Genomes Phase 3 data – to identify proxy variants (with LD $r^2 > 0.6$). We believe this approach maximises consistency with the predominant ancestry composition of the GWAS summary statistics and minimises discrepancies that could arise from LD structure differences across population.

We acknowledge that the ideal scenario would involve a reference panel perfectly matched to the ancestry composition of each GWAS dataset. However, given the strong predominance of European ancestry individuals in the datasets used for this analysis, we believe that any bias introduced by relying on the European reference panel was minimal. This clarification has been added to the **Methods** section of the revised manuscript.

Line 402:

*“**Material description.** We collected 45 publicly available GWAS summary statistics (**Supplementary Table 4**), spanning from 2017 to 09 June 2023. GWAS were selected if they included a large number of participants ($N \geq 10,000$) and a preference for datasets with diverse ancestral backgrounds, involving the majority of individuals of European ancestry, to enhance genetic diversity in our study. European-only studies were included when multi-ancestry data were unavailable.”*

Line 486:

*“**Clusters of pathogenetic processes.** We aggregated the 1,304 independent T2D-BP SNVs and investigated 45 endophenotypes GWAS summary statistics including T2D/BP (**Supplementary Table 4**) to cluster them into distinct groups based on pathogenetic processes. To reduce/avoid missingness across datasets, we identified high-LD proxies ($LD r^2 > 0.6$) using European ancestry reference panel from the 1000 Genomes Project Phase 3. Although some of the GWAS datasets were of multiple ancestries, we prioritised datasets comprising European-only studies or those with a high proportion of European ancestry participants. This approach mitigated potential discrepancies in LD patterns between the GWAS datasets and the reference panel.”*

Reviewer #3 (Remarks to the Author):

I read with great interest the manuscript by Pascat et al., which aims to elucidate the shared genetic architecture of type 2 diabetes (T2D) and hypertension by overlaying existing GWAS datasets for the two conditions and identifying clusters of overlapping variants. The authors further attempt to demonstrate that these clusters may enhance genetic risk prediction for the comorbidity. While the study is thoughtfully conceived and methodologically sound, several aspects merit further consideration and clarification (not ranked by importance):

1. Given the well-established comorbidity between T2D and hypertension—likely driven by shared environmental and biological risk factors, most notably obesity—it is perhaps unsurprising that a non-zero genetic correlation exists between the two (e.g., even if they were genetically “distinct” but shared 1 risk factor that was itself partly genetic, they would genetically correlate). This has already been demonstrated in prior studies. The manuscript might benefit from acknowledging this expected correlation more explicitly and discussing how their clustering approach adds value beyond confirming what is already previously studied.

We are thankful to the reviewer for this thoughtful comment. In response, we have revised the manuscript to explicitly acknowledge the well-established comorbidity and genetic correlation between T2D and high BP.

As rightly pointed out by the reviewer, the goal of the first part of our manuscript was to recapitulate and validate this known genetic relationship within the framework of our curated SNV set, providing context for the subsequent analyses, where the main added value of our study – the hierarchical clustering – moves beyond quantifying genetic correlation to decomposing the shared genetic architecture into biologically meaningful subgroups. As such, we amended the **Result** and **Discussion** sections.

Results

Line 90:

*“We performed linkage disequilibrium (LD) score regression using *ldsc*⁵³ and observed a positive genetic correlation between T2D and SBP ($r_g[SE]=0.25[0.028]$, $P=1.56 \times 10^{-19}$), DBP ($r_g[SE]=0.18[0.027]$, $P=1.38 \times 10^{-11}$), and PP ($r_g[SE]=0.23[0.029]$, $P=2.25 \times 10^{-15}$), in consensus with previous research.”*

Discussion

Line 329:

“In this large-scale multiomic study, we explored the complex genetic underpinnings of the comorbid relationship between T2D and high BP. To our knowledge, this is the first large-scale genetic analysis to dissect the shared genetic architecture of these two closely related metabolic conditions. Our analysis confirms and extends prior evidence of a positive genetic correlation and a large overlap in genetic signals between T2D and high BP^{54,55}. This observation is not unexpected, given the well-established comorbidity between T2D and hypertension – largely attributable to shared environmental and biological risk factors such as adiposity. Our approach extends beyond this by leveraging clustering of genetic variants to partition the T2D-BP genetic architecture into biologically coherent groups.”

2. It may be important for the authors to clarify the degree to which the respective GWAS datasets excluded individuals with the comorbid condition. For instance, if T2D GWAS cases include individuals with high blood pressure (which is probable), and vice versa, then the observed genetic overlap might in part reflect phenotypic correlations or sample contamination. A more detailed discussion—or at minimum, a caveat—regarding this potential bias may be warranted.

We thank the reviewer for raising this point. Upon careful examination of the methodologies used in the T2D GWAS by Mahajan et al. as well as the blood pressure GWAS by both Evangelou and Warren et al., we confirm that no explicit exclusion of individuals based on hypertension and high BP for T2D cases on one side, nor T2D status or high blood glucose for blood pressure traits on the other side was performed.

We acknowledge that no exclusion could introduce phenotypic correlations, potentially inflating the observed genetic correlations between T2D and BP traits. To mitigate the possibility of sample contamination bias, we performed our initial genetic correlation analyses using subsets of the GWAS that excluded the UK Biobank cohort – which is a major shared sample across these studies.

Additionally, we have now explicitly discussed this limitation in the revised manuscript, highlighting that while some degree of phenotypic correlation is inherent to these large datasets:

Line 373:

“We acknowledge several caveats in this work. The pathophysiological mechanisms involved in both T2D and high BP are not fully explained by genetics alone, and are influenced by variety of external and environmental factors, such as salt consumption or western diet. Moreover, the heterogeneity in sample origins, study designs, and sample sizes across the diverse included datasets could potentially reduce our ability to identify independent pathways. While the inclusion of individuals with potentially pre-existing comorbid conditions in GWAS studies of specific phenotypes, here T2D and BP, is often inevitable to capture the genetic architecture, it could somewhat inflate the estimated shared genetics due to phenotypic correlation. To mitigate the possibility of sample overlap, we used different subsets of T2D-BP GWASs for analyses that are sensitive to linkage disequilibrium structure. We mitigated the sample origin bias by prioritising datasets with multiple ancestries including European⁵⁶.”

3. The authors’ approach of selecting 1,401 genome-wide significant variants and clustering them before downstream annotation is reasonable. However, this top-hit strategy risks bias from the winner’s curse and excludes a large proportion of potentially informative polygenic signal. Would it not be more useful to use the full genome-wide summary statistics and apply methods such as GNOVA, partitioned LDSC, or some more sophisticated multivariate GWAS approach (e.g. genomic SEM) to partition the shared architecture more comprehensively? While I am not suggesting the current approach be abandoned, it would be helpful for the authors to justify their methodological choices and contrast them with possible alternatives.

We sincerely thank the reviewer for this comment. We agree that our top-hit strategy is a choice, and focussing on genome-wide significant variant may overlook a substantial proportion of the polygenic signal underlying complex traits. Methods such as GNOVA, partitioned LD score regression, or more advanced multivariate approaches like genomicSEM can indeed offer a complementary perspective on shared genetic architecture. For example, Park and colleagues recently applied a genomic SEM framework to dissect the genetic basis of Metabolic Syndrome.

We chose to focus on genome-wide significant loci for two main reasons. First, the clustering of genome-wide significant variants allows us to provide statistically robust and discrete groups of genetic signals, which are interpretable because they belong to different pathophysiological processes that have been discussed for individual conditions (e.g., higher adiposity, beta cell dysfunction). This approach prioritises biological interpretability based on the robustness of pre-existing evidence, which is important when investigating the genetic architecture of comorbid metabolic conditions such as T2D and BP, where pleiotropy and complex biological interactions prevail. In contrast, genome-wide methods, while statistically more comprehensive, often identify highly polygenic patterns that are harder to attribute to discrete biological mechanisms without additional follow-up.

Second, our primary goal was to generate tractable hypotheses about distinct mechanistic clusters driving T2D-BP comorbidity. By limiting our analysis to variants with the strongest evidence, we ensure that each cluster reflects robust, reproducible associations. We acknowledge that this approach inevitably excludes variants of moderate effect that contribute to the polygenic background, but it provides a clear starting point for biological characterisation and eventual experimental validation.

We acknowledge that a genome-wide multivariate framework such as genomic SEM could capture additional loci that did not reach genome-wide significance in either T2D or high BP alone. However, properly implementing it would require careful model specification (latent factors, sample overlap) and extensive biological interpretation, making it a project on its own. We are aware about the multivariate GWAS approaches, but the summary statistics are usually available for single trait GWAS, and the software tools are still to be adapted for the formatting of UKB files and respective UKB-based multivariate analyses. For this reason, we have not pursued a detailed genomic SEM of multivariate GWAS analysis within the scope of the present manuscript.

To explicitly acknowledge these methodological trade-offs, we have added the following paragraph to the limitation paragraph of the **Discussion**.

Line 386:

“Finally, we chose to focus on clustered genome-wide significant common variants to provide a clear and interpretable framework for dissecting the T2D-BP shared genetic relationships. While this strategy highlights robust mechanistic clusters, it may overlook a sizeable portion of the polygenic signal. Future studies could apply complementary approaches such as expanding this study to whole-exome or whole-genome datasets, or using multivariate GWAS or genomicSEM^{48,57} to further refine the understanding of T2D-BP relationship.”

4. The use of a “comorbidPGS” is intriguing. However, the authors should acknowledge that this directly reflects the genetic correlation already demonstrated

via LDSC. In other words, the predictive utility of one trait's PGS for the other trait reflects such genetic correlation. Although rescaling the LDSC genetic correlation into PGS's may offer some improved clinical utility, it would be helpful to frame these findings as extensions—rather than independent confirmations—of the same underlying genetic relationship.

We thank the reviewer for this point. We agree that the *comorbidPGS*³¹ approach reflects, at the individual level, the underlying genetic correlation that we initially quantified through LD score regression with *ldsc*^{53,58}. Accordingly, we have clarified in the manuscript that *comorbidPGS* analyses should be interpreted as an extension of the observed genetic correlations, rather than an independent replication.

Line 95:

“To further validate the LD score regression results, we constructed PGSs for T2D, SBP, DBP and PP in the UKB (Supplementary Table 1) using independent weights from GWASs (Methods). We probed whether a genetic predisposition towards one condition could predict the risk of the other using comorbidPGS³¹. T2D PGS was consistently associated with a modest increase in SBP, DBP, and PP (Beta_{PP}[SE]≥0.37[0.017] change in PP mmHg per one-unit increase in T2D PGS, P≤1.83×10⁻¹⁰⁶). SBP and PP PGSs were significantly associated with a higher risk of T2D (OR_{SBP}[95% CI]=1.07 [1.06-1.09] change in T2D odds per one-unit increase in SBP PGS, P=9.36×10⁻³⁵; OR_{PP}[95% CI]=1.07 [1.06-1.08], P=8.02×10⁻³¹). In contrast, DBP PGS had no impact on T2D risk (OR_{DBP}[95% CI]=1.01[0.999-1.02], P=0.062, Supplementary Table 2).”

Our rationale for using *comorbidPGS* lies in the latter part of the manuscript, where we used it to explore whether individuals with high scores in specific genetic clusters also exhibited distinct clinical outcomes. As pointed out by the reviewer, while LD score regression provides a population-level estimate, *comorbidPGS* offers a way to translate this relationship into individual risk profiles, which could eventually support clinical stratification strategies in the future.

5. The observation that cluster-informed PGSs predict comorbid outcomes is novel, but perhaps expected, given that these clusters were defined based on shared signal. To strengthen their conclusion that such partitioned clusters offer clinical value, the authors should compare these against the individual PGSs for T2D and BP, and ideally also a combined PGS (e.g., using genomic SEM-derived weights). Demonstrating that the cluster-based approach offers superior predictive accuracy over such approaches would substantiate the claim that their cluster-partitioned PGSs are more clinically useful.

We are thankful to the reviewer for this suggestion. To directly benchmark our cluster-partitioned PGSs against conventional single-trait scores, we used the already constructed in the first part of the results PGSs – one for T2D, and one for SBP – using a traditional pruning-and-thresholding (P+T) on the same discovery GWAS mentioned in the first part of the methods (without sample overlap of the UK Biobank).

We then compared the relative risk (RR) of T2D–BP comorbidity among individuals in the top decile and top tertile of each score. As expected, the T2D-weighted PGS achieved an RR of 1.60 in the top 10% (Table 1) and 1.40 in the top 33% (Table 2), and the SBP-weighted PGS

achieved an RR of 1.44 and 1.28, respectively, for their corresponding traits. In contrast, our combined unweighted cluster-partitioned PGS yielded substantially high comorbidity risk while similar to the weighted T2D PGS (RR = 2.13 in the top 10% and 1.62 in the top 33%). Therefore, we amended diligently the **Results** and the **Methods** sections of the manuscript to consistently correct the interpretation of our findings: these results are interesting because fewer SNPs combined in unweighted PGS can yield similar predictive ability to classic P+T PGS, and this paves the way for new tailored methods based on multiple PGS. We added these results in a new **Supplementary Table 14**.

Results

Line 273:

“Moreover, the individuals in the top 10% distribution of Metabolic Syndrome and Reduced beta-cells function combined PGSs, derived from 536 SNVs, showed a 2.13[1.96-2.31] fold increased risk of having comorbidity (Figure 4b, Supplementary Table 14), reaching the same RR as a traditional P+T weighted T2D PGS.”

Methods

Line 602:

“The summary table in Figure 4b provides the prevalence of individuals with T2D-BP comorbidity (of having both T2D and hypertension) based on being in the top decile (top 10%) of the partitioned unweighted PGS distribution. Additional results are available in Supplementary Table 14, showing both the top 10% and 33% of those partitioned PGS and comparing them with weighted T2D and SBP PGS.”

Additionally, we extended these comparisons to both **Results** and **Methods** sections by including the top tertiles of each weighted PGS alongside our cluster scores in the lifetime risk trajectories in a new **Supplementary Figure 11**. Individuals in the highest third of the T2D-weighted PGS aligned closely with those in the *Metabolic syndrome* cluster, and those in the highest third of the SBP-weighted PGS with the *Vascular dysfunction* cluster; however, the combined cluster PGS still produced an earlier and more pronounced separation of comorbidity risk curves. These results confirm the interest of cluster-partitioned PGS compared to single-trait PGSs (Figure 3)⁵⁹. We agree that deriving PGS weights via genomic SEM is a promising future direction, but implementation of such multivariate models extends beyond the scope of the current manuscript.

Results

Line 279:

“Survival analysis, using cumulative hazard plots, indicated that this elevated comorbidity risk was consistent and linear over 15 years of follow-up (year 0 representing the date of first diagnosis with either hypertension or T2D, Figure 4c, Supplementary Figure 11).”

Methods

Line 621:

“Figure 4c displays the cumulative hazard with 95% confidence intervals. Sensitivity analysis includes comparing this survival analysis using T2D weighted PGS and SBP weighted PGS respectively, and is included in Supplementary Figure 11.”

Table 1 : First part of the **Supplementary Table 14**: Summary table depicting the risk of T2D-BP comorbidity based on being in the top 10% of cluster PGSs and weighted PGS as controls in the UKB.

Top 10% of the PGS for	N individuals	T2D cases (%)	Hypertension cases (%)	T2D-BP comorbidity (%)	RR T2D	RR Hypertension	RR T2D-BP comorbidity
Weighted Pulse Pressure	45,925	7.90%	52.40%	6.30%	1.08	1.11	1.14
Weighted Diastolic Blood Pressure	458,489	7.30%	47.40%	5.50%	1.00	1.00	1.00
Weighted Systolic Blood Pressure	45,924	8.10%	58.40%	6.80%	1.11	1.23	1.24
Weighted T2D	45,925	15.90%	51.30%	11.80%	2.19	1.08	2.16
Reduced beta-cell function	45,906	11.30%	49.90%	8.50%	1.55	1.05	1.55
Vascular dysfunction	45,915	7.90%	52.90%	6.30%	1.08	1.12	1.14
Higher adiposity	45,913	9.80%	50.60%	7.50%	1.34	1.07	1.36
Metabolic Syndrome	45,891	10.10%	52.10%	7.90%	1.38	1.10	1.44
Inverse T2D-BP risk	45,895	8.10%	43.00%	5.70%	1.11	0.91	1.05
Metabolic Syndrome & Higher adiposity	4,605	13.10%	55.00%	10.20%	1.80	1.16	1.87
Metabolic Syndrome & Vascular dysfunction	4,670	11.00%	57.50%	8.90%	1.52	1.21	1.63
Metabolic Syndrome & Reduced beta-cell function	4,641	15.10%	54.60%	11.70%	2.07	1.15	2.13
Metabolic Syndrome & Higher adiposity & Reduced beta-cell function	470	18.90%	56.40%	14.30%	2.60	1.19	2.60
Whole cohort	459,247	7.30%	47.40%	5.50%	1.00	1.00	1.00

Table 2: Second part of the **Supplementary Table 14**, Summary table depicting the risk of T2D-BP comorbidity based on being in the top top 33% of cluster PGSs and weighted PGS as controls in the UKB.

Top 33% of the PGS for	N individuals	T2D cases (%)	Hypertension cases (%)	T2D-BP comorbidity (%)	RR T2D	RR Hypertension	RR T2D-BP comorbidity
Weighted Pulse Pressure	153,082	7.80%	50.50%	6.00%	1.07	1.07	1.10
Weighted Diastolic Blood Pressure	459,069	7.30%	47.40%	5.50%	1.00	1.00	1.00
Weighted Systolic Blood Pressure	153,082	7.80%	54.20%	6.30%	1.07	1.14	1.14
Weighted T2D	153,082	7.80%	54.20%	6.30%	1.07	1.14	1.60
Reduced beta-cell function	153,042	9.60%	48.80%	7.20%	1.32	1.03	1.31
Vascular dysfunction	153,068	7.70%	50.70%	6.00%	1.05	1.07	1.09

Higher adiposity	152,989	8.70%	49.40%	6.70%	1.20	1.04	1.22
Metabolic Syndrome	153,080	9.00%	50.50%	6.90%	1.23	1.07	1.26
Inverse T2D-BP risk	153,037	7.80%	44.50%	5.60%	1.07	0.94	1.03
Metabolic Syndrome & Higher adiposity	51,481	10.50%	52.30%	8.20%	1.45	1.10	1.49
Metabolic Syndrome & Vascular dysfunction	51,446	9.30%	53.60%	7.40%	1.28	1.13	1.35
Metabolic Syndrome & Reduced beta-cell function	51,446	9.30%	53.60%	7.40%	1.28	1.13	1.62
Metabolic Syndrome & Higher adiposity & Reduced beta-cell function	17,605	13.50%	53.80%	10.50%	1.86	1.14	1.92
Whole cohort	459,247	7.30%	47.40%	5.50%	1.00	1.00	1.00

Figure 3 : Cumulative hazard plot of T2D-BP comorbidity stratified by being in the top 33 percent of the unweighted partitioned PGS and T2D/SBP weighted PGS. This figure is included in the supplementary materials as **Supplementary Figure 11**.

6. The statement that their approach enables “identifying younger individuals at high risk more effectively” is difficult to support using UKB, which is limited to middle-aged and older adults. A more cautious phrasing is advised, unless the authors are able to validate their findings in younger cohorts (e.g., birth cohorts with longitudinal follow-up). The current data do not allow conclusions about predictive performance in truly young populations. Furthermore, the existing GWAS data is primarily from late-

adult/mid-adult populations, but the ongoing “lifecourse GWAS” may help answer such questions in the future.

We are thankful to Reviewer for this point and fully agree about the need for precise terminology. We also agree that UK biobank youngest age at enrolment of participants was 37 y.o., e.g. adult age. We therefore have amended the three mentions of “young(er) individuals” in the **Results** and **Discussion** sections.

Results

Line 280:

*“This suggests that individuals with high PGS distributions remain at increased risk of comorbidity throughout their life course. **Consequently, partitioned PGSs could bear significant predictive ability to identify high-risk individuals at an earlier age^{60,61}.**”*

Line 324:

*“The unweighted PGS effectively delineated the differences among the T2D-BP cluster SNVs. **The partitioning of PGSs in T2D-BP comorbidity highlighted that a reduced number of SNVs can predict multiple related conditions through diverse pathophysiological processes.**”*

Discussion

Line 363:

*“We bring forward a property of partitioned PGS to differentially predict related T2D – high BP conditions⁶². While PGS has shown predictive value for an expanding array of common diseases, such as for instance CAD^{63,64}, its clinical application remains limited due to a range of pitfalls⁶⁵. **In this study, we report how specific clusters, particularly the Metabolic Syndrome and Reduced beta-cell function clusters, can identify a sub-population of individuals over twice the general population risk of T2D-BP comorbidity.**”*

Additionally, we modified accordingly the conclusion paragraph of the **Discussion**:

Line 393:

*“In this study, we highlight the mechanistic heterogeneity underlying T2D and high BP, **demonstrating that their genetic relationship is not driven by a single shared pathophysiological process but instead arises from five distinct pathways contributing to the T2D-BP comorbidity. The partitioned PGSs, derived through clustering, enables modelling of differential lifetime risk trajectories associated with T2D-BP comorbidity. These findings provide a framework for future investigations into stratified risk prediction and pathophysiology-informed approaches to manage comorbid conditions.**”*

7. The manuscript references the mitigation of “... pleiotropy effects” (row 329) but the meaning of this is unclear. If the authors refer to horizontal pleiotropy between T2D and BP, it is not evident how their method—especially one that selects top variants—would control for this. It is also not clear how such pleiotropy would affect their findings (if at all??). Some clarification or rewording is needed to prevent conceptual ambiguity.

We thank the reviewer for pointing out this misleading phrase. We do not aim to eliminate pleiotropy per se – indeed, the presence of shared genetic effects between T2D and high blood pressure is central to our study of comorbidity. Rather, our intent was to reduce broad cross-trait associations driven by variants affecting multiple traits through independent mechanisms, which could confound the clustering.

Spencer et al. recently highlighted, in a large-scale preprint, that genome-wide significant variants tend to capture trait-specific biology more robustly than sub-threshold signals, particularly when comparing loci across related traits⁵². Their findings support the notion that top GWAS hits are more likely to be enriched for causal variants with interpretable, trait-specific effects. This observation builds on prior work by Watanabe et al. and Qi et al., which showed that genome-wide significant loci exhibit greater tissue and pathway specificity, and are less likely to reflect broad, non-specific pleiotropy compared to variants with weaker associations^{66,67}. Together, these studies suggest that focusing on genome-wide significant variants increases the likelihood of capturing biologically coherent and trait-relevant signals.

To clarify this point, we do not cite anymore “pleiotropy effects” and revised the relevant passage in the **Discussion**.

Line 338:

“We curated a set of genome-wide significant common SNVs associated with T2D and high BP, thereby enriching for variants with stronger and trait-specific effects, and reducing the influence of broader, less specific cross-trait associations^{52,67}.”

8. The identification of a cluster associated with “Inverse T2D-BP risk” is certainly intriguing. However, such a pattern could emerge naturally given incomplete overlap in genetic architectures. Mathematically, one would expect such clusters to exist in the symmetric difference between genetic variance of each condition. While still important to highlight, the biological interpretation should be handled cautiously.

We thank the reviewer for this thoughtful observation. We agree that the emergence of an “Inverse T2D-BP risk” cluster may, in part, reflect the properties of partially overlapping genetic architectures. As the reviewer correctly notes, variants uniquely associated with one trait but not the other – or with opposite effect directions – are expected to appear in the symmetric difference between the two trait-specific genetic architectures.

In response, we have revised the **Discussion** section of the manuscript to temper the biological interpretation of this cluster and to explicitly acknowledge that such a pattern may arise from statistical, rather than purely biological, considerations.

Line 346:

“We discovered an intriguing cluster of variants with an Inverse T2D-BP risk profile, implicating retinol metabolism. Although recent meta-analysis on the role of retinol role in T2D and BP regulation have yielded inconsistent results, one retinol derivative, retinoic acid, has consistently been linked to higher IR and reduced cardiovascular events⁶⁸. While we propose a mechanistically plausible pathophysiological hypothesis for inverse the T2D-BP risk effects, these patterns may also arise from incomplete overlap in the genetic architectures underlying T2D and BP.”

9. The authors may wish to acknowledge more explicitly that their findings are limited to common variants. Extending this clustering framework to include whole-exome or whole-genome—should such data become available—could offer additional insights and deserves mention as a future direction.

We thank the reviewer for this suggestion. We have now added a paragraph in the **Discussion** section to acknowledge this limitation and highlight the potential of integrating whole-exome or whole-genome data as a future direction to enrich the biological resolution of this framework.

Line 386:

“Finally, we chose to focus on clustered genome-wide significant common variants to provide a clear and interpretable framework for dissecting the T2D-BP shared genetic relationships. While this strategy highlights robust mechanistic clusters, it may overlook a sizeable portion of the polygenic signal. Future studies could apply complementary approaches such as expanding this study to whole-exome or whole-genome datasets, or using multivariate GWAS or genomicSEM^{48,57} to further refine the understanding of T2D-BP relationship.”

10. The conclusions in the final sections overstate the demonstrated findings. In particular, the claim of “improved accuracy” and the suggestion that this work provides “a foundation for precision medicine” are premature without direct comparison to conventional PGS strategies (the individual ones or a combined PGS, see point 5) or demonstration of direct clinical utility. A more tempered conclusion would better reflect the limitations and incremental nature of this work.

We thank the reviewer for this comment and agree that the initial phrasing may have overstated the implications of our findings. We have now revised the final paragraph of the Discussion to adopt a more measured tone and better reflect the exploratory nature of our work. Specifically, we removed claims related to “improved accuracy” and “precision medicine” and reframed our findings as hypothesis-generating.

Line 393:

“In this study, we highlight the mechanistic heterogeneity underlying T2D and high BP, demonstrating that their genetic relationship is not driven by a single shared pathophysiological process but instead arises from five distinct pathways contributing to the T2D-BP comorbidity. The partitioned PGSs, derived through clustering, enable modelling of differential lifetime risk trajectories associated with T2D-BP comorbidity. These findings provide a framework for future investigations into stratified risk prediction and pathophysiology-informed approaches to manage comorbid conditions.”

11. I am somewhat concerned about the use of GWASs data from different ancestries. While I share the authors enthusiasm to increase diversity in data, most methods employed assume a fixed or common LD structure – and combining data from across ancestries would therefore introduce bias via population stratification. Please

consider the implications of this and amend accordingly. I note that the authors correctly restricted their LDSC to European ancestry.

We are thankful to the reviewer for this point. To balance, power, generalisability and methodological rigour, we implemented a two-stage strategy. We minimised the impact of ancestry-related heterogeneity for LD score and PGS analysis, as described in Reviewer 1's comments 1)a)b)c) and Reviewer 2's comments 3)4). However, for downstream clustering and functional annotation, we employed larger, more powerful multi-ancestry GWASs to ensure sufficient variant discovery and statistical power. In doing so, we took care to select studies in which European ancestry individuals comprised the majority of the sample.

When index variants were missing from certain datasets, proxies were identified using LD patterns derived from the European subset of the 1000 Genomes reference panel. While this ensured consistency across analyses, we recognise that ancestry-specific LD variation may limit the precision of proxy-based mapping in a multi-ancestry setting. We have now clarified this methodological compromise in the manuscript as described in Reviewer 2's last comment.

Line 403:

“GWAS were selected if they included a large number of participants ($N \geq 10,000$) and a preference for datasets with diverse ancestral backgrounds, involving the majority of individuals of European ancestry, to enhance genetic diversity in our study.”

Line 488:

“To reduce/avoid missingness across datasets, we identified high-LD proxies ($LD r^2 > 0.6$) using European ancestry reference panel from the 1000 Genomes Project Phase 3. Although some of the GWAS datasets were of multiple ancestries, we prioritised datasets comprising European-only studies or those with a high proportion of European ancestry participants. This approach mitigated potential discrepancies in LD patterns between the GWAS datasets and the reference panel.”

Overall, this is an ambitious and promising study. With some refinement and additional contextualization, it could make a valuable contribution to the literature on polygenic architecture and risk prediction for comorbid cardiometabolic diseases.

Reference

1. Mahajan, A. *et al.* Fine-mapping type 2 diabetes loci to single-variant resolution using high-density imputation and islet-specific epigenome maps. *Nat Genet* **50**, 1505–1513 (2018).
2. Warren, H. R. *et al.* Genome-wide association analysis identifies novel blood pressure loci and offers biological insights into cardiovascular risk. *Nat Genet* **49**, 403–415 (2017).
3. Evangelou, E. *et al.* Genetic analysis of over 1 million people identifies 535 new loci associated with blood pressure traits. *Nat Genet* **50**, 1412–1425 (2018).
4. Suzuki, K. *et al.* Genetic drivers of heterogeneity in type 2 diabetes pathophysiology. *Nature* **627**, 347–357 (2024).
5. Keaton, J. M. *et al.* Genome-wide analysis in over 1 million individuals of European ancestry yields improved polygenic risk scores for blood pressure traits. *Nat Genet* **56**, 778–791 (2024).
6. Purcell, S. *et al.* PLINK: A tool set for whole-genome association and population-based linkage analyses. *Am J Hum Genet* **81**, 559–575 (2007).
7. Vujkovic, M. *et al.* Discovery of 318 new risk loci for type 2 diabetes and related vascular outcomes among 1.4 million participants in a multi-ethnic meta-analysis. **52**, 680–691 (2020).
8. Morris, A. P. *et al.* Large-scale association analysis provides insights into the genetic architecture and pathophysiology of type 2 diabetes. *Nat Genet* **44**, 981–990 (2012).
9. Mahajan, A. *et al.* Refining the accuracy of validated target identification through coding variant fine-mapping in type 2 diabetes. *Nat Genet* **50**, 559–571 (2018).
10. Saxena, R. *et al.* Genome-Wide Association Study Identifies a Novel Locus Contributing to Type 2 Diabetes Susceptibility in Sikhs of Punjabi Origin From India. *Diabetes* **62**, 1746–1755 (2013).
11. Gaulton, K. J. *et al.* Genetic fine mapping and genomic annotation defines causal mechanisms at type 2 diabetes susceptibility loci. *Nat Genet* **47**, 1415–1425 (2015).
12. Cho, Y. S. *et al.* Meta-analysis of genome-wide association studies identifies eight new loci for type 2 diabetes in east Asians. *Nat Genet* **44**, 67–72 (2012).
13. Tragante, V. *et al.* Gene-centric Meta-analysis in 87,736 Individuals of European Ancestry Identifies Multiple Blood-Pressure-Related Loci. *The American Journal of Human Genetics* **94**, 349–360 (2014).
14. Liang, J. *et al.* Single-trait and multi-trait genome-wide association analyses identify novel loci for blood pressure in African-ancestry populations. *PLoS Genet* **13**, e1006728 (2017).
15. Levy, D. *et al.* Genome-wide association study of blood pressure and hypertension. *Nat Genet* **41**, 677–687 (2009).
16. Johnson, A. D. *et al.* Association of Hypertension Drug Target Genes With Blood Pressure and Hypertension in 86 588 Individuals. *Hypertension* **57**, 903–910 (2011).

17. Takeuchi, F. *et al.* Interethnic analyses of blood pressure loci in populations of East Asian and European descent. *Nat Commun* **9**, 5052 (2018).
18. Franceschini, N. *et al.* Genome-wide Association Analysis of Blood-Pressure Traits in African-Ancestry Individuals Reveals Common Associated Genes in African and Non-African Populations. *The American Journal of Human Genetics* **93**, 545–554 (2013).
19. Genetic variants in novel pathways influence blood pressure and cardiovascular disease risk. *Nature* **478**, 103–109 (2011).
20. Giri, A. *et al.* Trans-ethnic association study of blood pressure determinants in over 750,000 individuals. *Nat Genet* **51**, 51–62 (2019).
21. Kato, N. *et al.* Trans-ancestry genome-wide association study identifies 12 genetic loci influencing blood pressure and implicates a role for DNA methylation. *Nat Genet* **47**, 1282–1293 (2015).
22. Wang, Y. *et al.* Whole-genome association study identifies *STK39* as a hypertension susceptibility gene. *Proceedings of the National Academy of Sciences* **106**, 226–231 (2009).
23. Hoffmann, T. J. *et al.* Genome-wide association analyses using electronic health records identify new loci influencing blood pressure variation. *Nat Genet* **49**, 54–64 (2017).
24. Liu, C. *et al.* Meta-analysis identifies common and rare variants influencing blood pressure and overlapping with metabolic trait loci. *Nat Genet* **48**, 1162–1170 (2016).
25. Wain, L. V. *et al.* Novel Blood Pressure Locus and Gene Discovery Using Genome-Wide Association Study and Expression Data Sets From Blood and the Kidney. *Hypertension* **70**, (2017).
26. Foley, C. N., Mason, A. M., Kirk, P. D. W. & Burgess, S. MR-Clust: clustering of genetic variants in Mendelian randomization with similar causal estimates. *Bioinformatics* **37**, 531–541 (2021).
27. Foley, C. N., Mason, A. M., Kirk, P. D. W. & Burgess, S. MR-Clust: clustering of genetic variants in Mendelian randomization with similar causal estimates. *Bioinformatics* **37**, 531–541 (2021).
28. Udler, M. S. *et al.* Type 2 diabetes genetic loci informed by multi-trait associations point to disease mechanisms and subtypes: A soft clustering analysis. *PLoS Med* **15**, (2018).
29. Vaura, F. *et al.* Multi-Trait Genetic Analysis Reveals Clinically Interpretable Hypertension Subtypes. *Circ Genom Precis Med* **15**, e003583 (2022).
30. Smith, K. *et al.* Multi-ancestry polygenic mechanisms of type 2 diabetes. *Nat Med* **30**, 1065–1074 (2024).
31. Pascat, V. *et al.* comorbidPGS: an R package assessing shared predisposition between Phenotypes using Polygenic Scores. *Hum Hered* (2024) doi:10.1159/000539325.
32. Privé, F., Arbel, J. & Vilhjálmsón, B. J. LDpred2: better, faster, stronger. *Bioinformatics* **36**, 5424–5431 (2021).

33. Choi, S. W., Mak, T. S. H. & O'Reilly, P. F. Tutorial: a guide to performing polygenic risk score analyses. *Nat Protoc* **15**, 2759–2772 (2020).
34. Hippisley-Cox, J. & Coupland, C. Development and validation of QDiabetes-2018 risk prediction algorithm to estimate future risk of type 2 diabetes: cohort study. *BMJ* j5019 (2017) doi:10.1136/bmj.j5019.
35. Hippisley-Cox, J., Coupland, C. & Brindle, P. Development and validation of QRISK3 risk prediction algorithms to estimate future risk of cardiovascular disease: prospective cohort study. *BMJ* j2099 (2017) doi:10.1136/bmj.j2099.
36. Fuat, A. *et al.* A polygenic risk score added to a QRISK®2 cardiovascular disease risk calculator demonstrated robust clinical acceptance and clinical utility in the primary care setting. *Eur J Prev Cardiol* **31**, 716–722 (2024).
37. *The IDF Consensus Worldwide Definition of the METABOLIC SYNDROME.* <https://idf.org/media/uploads/2023/05/attachments-30.pdf> (2023).
38. Fall, T. *et al.* The Role of Adiposity in Cardiometabolic Traits: A Mendelian Randomization Analysis. *PLoS Med* **10**, e1001474 (2013).
39. Holmes, M. V. *et al.* Causal Effects of Body Mass Index on Cardiometabolic Traits and Events: A Mendelian Randomization Analysis. *The American Journal of Human Genetics* **94**, 198–208 (2014).
40. Prasad, R. *et al.* Dual Metrics of Obesity: Evaluating BMI and Central Obesity as Indicators of Cardiometabolic Risk in Rural India. Preprint at <https://doi.org/10.1101/2024.08.13.24311925> (2024).
41. Ren, H., Guo, Y., Wang, D., Kang, X. & Yuan, G. Association of normal-weight central obesity with hypertension: a cross-sectional study from the China health and nutrition survey. *BMC Cardiovasc Disord* **23**, 120 (2023).
42. Adams, B., Jacocks, L. & Guo, H. Higher BMI is linked to an increased risk of heart attacks in European adults: a Mendelian randomisation study. *BMC Cardiovasc Disord* **20**, 258 (2020).
43. Karhunen, V. *et al.* The interplay between inflammatory cytokines and cardiometabolic disease: bi-directional mendelian randomisation study. *BMJ Medicine* **2**, e000157 (2023).
44. Gill, D. *et al.* Risk factors mediating the effect of body mass index and waist-to-hip ratio on cardiovascular outcomes: Mendelian randomization analysis. *Int J Obes* **45**, 1428–1438 (2021).
45. Vasiliki, L. *et al.* Random glucose GWAS in 493,036 individuals provides insights into diabetes pathophysiology, complications and treatment stratification. *Unnur Thorsteinsdottir* **6**, 82.
46. Huang, T. *et al.* Genetic Predisposition to Central Obesity and Risk of Type 2 Diabetes: Two Independent Cohort Studies. *Diabetes Care* **38**, 1306–1311 (2015).
47. Laaksonen, D. *et al.* Sex hormones, inflammation and the metabolic syndrome: a population-based study. *Eur J Endocrinol* 601–608 (2003) doi:10.1530/eje.0.1490601.

48. Park, S. *et al.* Multivariate genomic analysis of 5 million people elucidates the genetic architecture of shared components of the metabolic syndrome. *Nat Genet* **56**, 2380–2391 (2024).
49. Després, J.-P. & Lemieux, I. Abdominal obesity and metabolic syndrome. *Nature* **444**, 881–887 (2006).
50. Huang, P. L. A comprehensive definition for metabolic syndrome. *DMM Disease Models and Mechanisms* **2**, 231–237 (2009).
51. Durinck, S., Spellman, P. T., Birney, E. & Huber, W. Mapping identifiers for the integration of genomic datasets with the R/Bioconductor package biomaRt. *Nat Protoc* **4**, 1184–1191 (2009).
52. Spence, J. P. *et al.* Specificity, length, and luck: How genes are prioritized by rare and common variant association studies. Preprint at <https://doi.org/10.1101/2024.12.12.628073> (2024).
53. Bulik-Sullivan, B. *et al.* LD score regression distinguishes confounding from polygenicity in genome-wide association studies. *Nat Genet* **47**, 291–295 (2015).
54. Wielscher, M. *et al.* Genetic correlation and causal relationships between cardio-metabolic traits and lung function impairment. *Genome Med* **13**, 104 (2021).
55. Vattikuti, S., Guo, J. & Chow, C. C. Heritability and Genetic Correlations Explained by Common SNPs for Metabolic Syndrome Traits. *PLoS Genet* **8**, e1002637 (2012).
56. Graham, S. E. *et al.* The power of genetic diversity in genome-wide association studies of lipids. *Nature* **600**, 675–679 (2021).
57. Grotzinger, A. D. *et al.* Genomic structural equation modelling provides insights into the multivariate genetic architecture of complex traits. *Nat Hum Behav* **3**, 513–525 (2019).
58. Bulik-Sullivan, B. *et al.* An atlas of genetic correlations across human diseases and traits. *Nat Genet* **47**, 1236–1241 (2015).
59. Klau, J. H. *et al.* AI-based multi-PRS models outperform classical single-PRS models. *Front Genet* **14**, (2023).
60. Jiang, X., Holmes, C. & McVean, G. The impact of age on genetic risk for common diseases. *PLoS Genet* **17**, e1009723 (2021).
61. Thompson, D. J. *et al.* UK Biobank release and systematic evaluation of optimised polygenic risk scores for 53 diseases and quantitative traits. doi:10.1101/2022.06.16.22276246.
62. Lambert, S. A., Abraham, G. & Inouye, M. Towards clinical utility of polygenic risk scores. *Hum Mol Genet* **28**, R133–R142 (2019).
63. Surakka, I. *et al.* Sex-Specific Survival Bias and Interaction Modeling in Coronary Artery Disease Risk Prediction. *Circ Genom Precis Med* **16**, e003542 (2023).
64. Ma, Y. & Zhou, X. Genetic prediction of complex traits with polygenic scores: a statistical review. *Trends in Genetics* **37**, 995–1011 (2021).
65. Novembre, J. *et al.* Addressing the challenges of polygenic scores in human genetic research. *The American Journal of Human Genetics* **109**, 2095–2100 (2022).

66. Qi, G. *et al.* Genome-wide large-scale multi-trait analysis characterizes global patterns of pleiotropy and unique trait-specific variants. *Nat Commun* **15**, 6985 (2024).
67. Watanabe, K. *et al.* A global overview of pleiotropy and genetic architecture in complex traits. *Nat Genet* **51**, 1339–1348 (2019).
68. Olsen, T. & Blomhoff, R. Retinol, Retinoic Acid, and Retinol-Binding Protein 4 are Differentially Associated with Cardiovascular Disease, Type 2 Diabetes, and Obesity: An Overview of Human Studies. *Advances in Nutrition* **11**, 644–666 (2020).

AUTHOR REBUTTAL LETTER

Dear Editor and Reviewers,

We are very grateful to the reviewers, editor and whole editorial team for their time and thoughtful suggestions.

We carefully addressed the reviewer's remaining comments which we believe have substantially improved our manuscript. Additionally, we diligently covered all the points from the editors' author guidance table.

In the below document we provide responses to the review comments point-by-point, using **bold blue text** to highlight the reviewer's question and plain black text for our response. We also provide citations of the introduced changes in *italic text, with added words in red*.

We **highlighted in green** the changes introduced to the main text and followed other instructions for the submission.

Reviewer Comments

Reviewer #1 (Remarks to the Author):

Previous comment 3 suggesting a comparison with previously reported BP clusters. The authors have dealt with this comment well. However, I wonder if it would be better to present to comparisons between clusters in Supplementary Figure 6 (and also in Supplementary Figure 5) using a Sankey plot. I believe this more clearly shows how the clusters are related to each other.

We thank the reviewer for this helpful suggestion. We have modified **Supplementary Figures 5** and **6** to include the corresponding Sankey plots, now presented as subfigures 5b and 6c-d, which we believe illustrate more clearly the relationships between the clusters.

Figure 1: Updated Supplementary Figure 5

Figure 2: Updated Supplementary Figure 6

Previous comment 5 about the survival analysis. I accept the authors argument here, but I wonder if this should be emphasized in the main text (maybe discussion) as the points they have made are important.

We thank the reviewer for this valuable comment. We have added the following paragraph to the Discussion section to emphasise the key points raised in our response from the first rebuttal letter regarding the survival analysis:

Line 372:

For the survival analysis, individuals were assigned to the cluster corresponding to their highest partitioned PGS, to emulate a potential clinical framework where patients could be stratified into their predominant mechanistic risk pathway for targeted prevention. While this approach simplifies the genetic architecture, it illustrates how refined, cluster-specific PGSs could ultimately complement existing clinical tools such as QDiabetes and QRisk to enable earlier, pathway-informed interventions.